# Classification of Synoptic Conditions of Summer Floods in Polish Sudeten Mountains

**Ewa Bednorz** [1],*, **Dariusz Wrzesiński** [2], **Arkadiusz M. Tomczyk** [1] and **Dominika Jasik** [2]

[1]  Department of Climatology, Adam Mickiewicz University, 61-680 Poznań, Poland
[2]  Department of Hydrology and Water Management, Adam Mickiewicz University, 61-680 Poznań, Poland
*  Correspondence: ewabedno@amu.edu.pl; Tel.: +48-61-829-6267

**Abstract:** Atmospheric processes leading to extreme floods in the Polish Sudeten Mountains were described in this study. A direct impact of heavy precipitation on extremely high runoff episodes was confirmed, and an essential role of synoptic conditions in triggering abundant rainfall was proved. Synoptic conditions preceding each flood event were taken into consideration and the evolution of the pressure field as well as the moisture transport was investigated using the anomaly-based method. Maps of anomalies, constructed for the days prior to floods, enabled recognizing an early formation of negative centers of sea level pressure and also allowed distinguishing areas of positive departures of precipitable water content over Europe. Five cyclonic circulation patterns of different origin, and various extent and intensity, responsible for heavy, flood-triggering precipitation in the Sudetes, were assigned. Most rain-bringing cyclones form over the Mediterranean Sea and some of them over the Atlantic Ocean. A meridional southern transport of moisture was identified in most of the analyzed cases of floods. Recognizing the specific meteorological mechanisms of precipitation enhancement, involving evolution of pressure patterns, change in atmospheric moisture and occurrence of precipitation may contribute to a better understanding of the atmospheric forcing of floods in mountain areas and to improve predicting thereof.

**Keywords:** Sudeten Mountains; flood; extreme precipitation; atmospheric circulation; synoptic conditions; sea level pressure; cyclone tracks; precipitable water

## 1. Introduction

Floods are considered to be one of the most common and destructive natural hazards. They represent a significant risk to people and can have a major economic impact on society. Although Mudelsee et al. [1,2] reveals that observations from Europe do not show a clear increase in flood occurrence rate, extreme river floods still have had devastating effects in central Europe in recent years. Despite improved early warning systems and better flood control infrastructure, flood damage has grown considerably, as the flood-vulnerable areas have gained richer infrastructure and wealth [3,4]. While long time data series do not show an upward trend of flood occurrence [1,5,6], there is widespread concern that not only flood damage but also flood hazards could be on the rise. This is partly justified by most of the model-based projections which simulate future increase of flood risk in central Europe, based on an assumption of increasing precipitation due to higher content of water vapor in the warmer air [3,7,8].

Among meteorological conditions which trigger the advent of flood, precipitation is considered as the most significant driver [9]. In contrast to droughts, which result from long-term insufficient receiving of precipitation, floods are rather rapid phenomena that originate from extremely high rainfall rates, and high precipitation efficiency [10]. Because of relevance to natural hazards, including

floods, the analysis of trends and variations in extreme precipitation has recently received a great deal of attention [11].

The combination of high atmospheric moisture content, rainfall rates and precipitation efficiency resulting in extreme floods is usually achieved through distinct synoptic-scale circulation patterns [12–15]. Additionally, the interaction with the terrain landforms gains crucial significance, particularly within small catchments in mountain areas [16–19]. As the extreme flood events are rare, it is difficult to define their general synoptic conditions based on a large number of events. Nonetheless, the analysis of several carefully selected and well-documented flood events can provide the range of relevant flood-producing synoptic patterns and may enable drawing conclusions concerning atmospheric mechanisms leading to these extreme events.

Some previous studies on synoptic conditions of floods in the southern, mountainous part of Poland concentrated rather on the Carpathian Mountains and they referred mostly to macroscale circulation patterns leading to heavy-rain events [17,20–22]. Several case studies concerning the Sudeten Mountains were dedicated to particularly damaging flood events, like for example in July 1970 [23], July 1997 [24,25] or May–June 2010 [25,26].

The aim of this study was to identify the synoptic patterns that control precipitation and flooding in three catchments located in the Polish Sudeten Mountains. To this end, a detailed analysis of 17 case studies of the largest recorded flood events was undertaken. Because of the delayed reaction of catchment runoff, the synoptic conditions preceding each flood event were taken into consideration. This enabled the storm track determination of rain-bringing cyclones, to assign trajectories of humid air masses and to indicate their source regions. Thus, specific meteorological flood-triggering mechanisms, involving evolution of pressure patterns, changing in atmospheric moisture and occurrence of precipitation, which lead to extreme floods in the Polish Sudeten, were detected and described.

## 2. Area of the Study

The Sudeten Mountains and Sudeten foreland are the second richest in water resource regions of Poland after the Carpathians. At the same time they are the most flood-prone areas in the country, with floods recorded particularly during the summer months. The study area is located in the Odra River Basin, which belongs to the Baltic Sea drainage basin (Figure 1). Its hydrographic system is diversified due to a varied terrain and geology. River gorges located within the foundations of older subsoil and the concentric river network in the Jeleniogórska and Kłodzka Valleys provide favorable conditions for formation of high-water stages and floods. The slope of riverbeds in mountain valleys exceed 5‰ and, locally in the highest sections, reaches 1% or more [27].

The main characteristics of the Sudeten rivers include abundance of their water resources, turbulent water flow, rapid response to changes in water supply in the upper parts of their catchments, and large fluctuations in water stages in the yearly cycle. The hydrological regime of these rivers determines the occurrence of high-water stages in different seasons. However, the most violent and dangerous are summer floods, during which the rivers, and especially the mountain streams, carry even several dozen times more water than on the average, as for example the Biała Lądecka River in Lądek Zdrój, whose discharge during the flood in July 1997 was 160 times larger than its average long-term discharge [28].

The prolonged and regional-scale rainfall is one of the most common causes of flooding. During such events the ground retention capacity is exceeded and the surface runoff is increased. About 75% of high-water events recorded in the Odra River occurs as a result of the prolonged and also intense rains from June to August [29]. The Jizera Mountains and the Śnieżnik Massif belong to areas most exposed to high-water events caused by rainstorms [30].

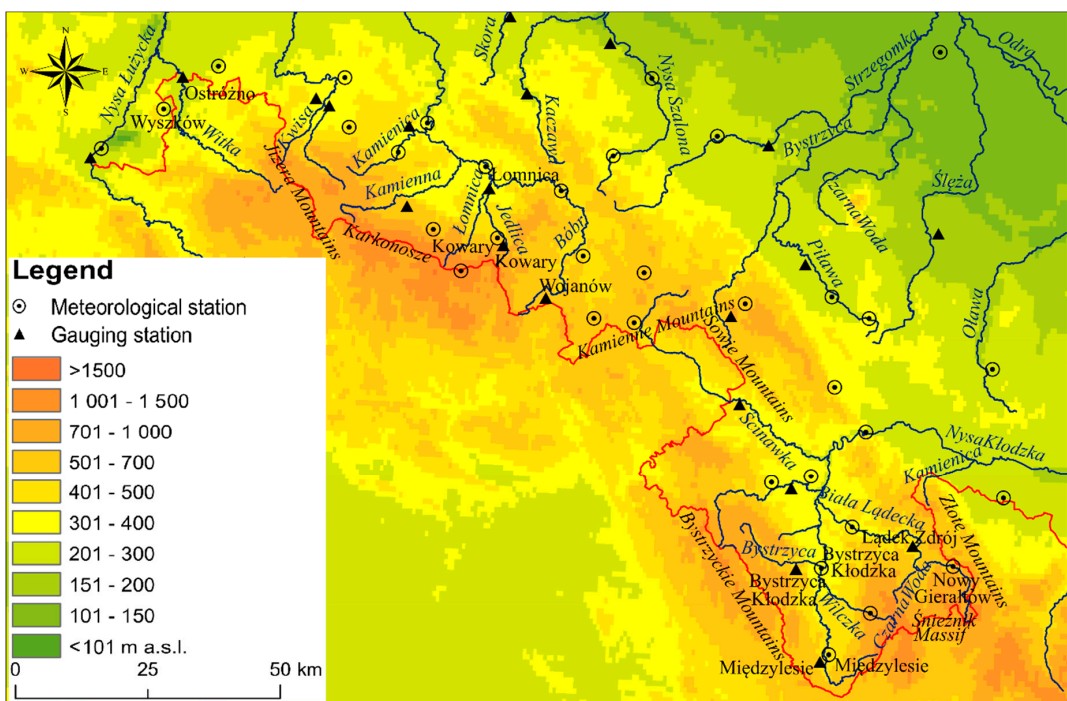

**Figure 1.** Area of the study with location of 21 gauging stations and 43 meteorological stations used in the study.

## 3. Data and Methods

In this study the upper sections of 21 rivers drained by six tributaries of the Odra River, located in Sudeten Mountains, were analyzed. Among them the largest and most water-rich are the Nysa Kłodzka and the Bóbr, which are also major rivers draining the Sudeten Mountains. On each of these rivers in the flood periods—during which the flow maxima were higher than the mean-high discharge (MHQ)—of the years 1971–2015 were distinguished. To this end, daily discharges recorded at 21 gauging stations located on rivers in the Polish part of the Sudetes in the multi-year period 1971–2015 were analyzed (Figure 1). The data come from the Institute of Meteorology and Water Resources Management (IMGW-PIB) databases (available at https://dane.imgw.pl/data/). All IMGW-PIB data undergo quality control both manual and automatic. For further analyzes, only floods that occurred simultaneously on 60% of the examined rivers were selected. From the same source (IMGW-PIB) daily precipitation totals for 43 meteorological measurement stations were obtained and on this basis maps of weekly totals for seven days preceding each flood event (including the flood culmination day) were constructed.

In the synoptic climatology, whose domain is an analysis of the occurrence of some environmental phenomena from the perspective of atmospheric circulation, one of the two fundamental approaches can be adopted: 'circulation to environment' or 'environment to circulation'. In the first approach, an atmospheric circulation classification is performed first and then circulation types are related to an environmental phenomenon, and in the second method, the circulation classification is carried along a specific environment-based criteria set for a particular environmental phenomenon—flood, in the case of this study [31–33]. The 'environment to circulation' approach was applied in this study and after selecting the flood events a composite characteristic of synoptic conditions was performed.

To provide a broad atmospheric background of the chosen flood events mean daily sea level pressure (SLP) data were acquired in order to create maps of the pressure fields and to identify circulation patterns. Analysis of the low level baric conditions were supported by the height of the 500 hPa geopotential level (Z500). SLP and Z500 data were selected from the National Centre for Environmental Prediction/National Centre for Atmospheric Research (NCEP/NCAR) reanalysis

data [34]. Besides SLP, daily total precipitable water (PW) data were collected from the same resources. Precipitable water which can be defined as the total water that is contained in a column of unit cross section extending all of the way from the earth's surface to the "top" of the atmosphere. Mathematically, if *x(p)* is the mixing ratio at the pressure level, *p*, then the precipitable water vapor, *W*, contained in a layer bounded by pressures $p_1$ and $p_2$ is given by the equation [35]:

$$W = \frac{1}{\rho g} \int_{p_2}^{p_1} x dp \tag{1}$$

Thus, PW is a variable that expresses the total content of water in the atmosphere and there is a general correlation between precipitation amounts in given storms and the PW of the air masses involved in those storms. Both SLP and PW reanalysis data used in this study are grid-based with a resolution $2.5 \times 2.5°$. Such a spatial grid resolution is sufficient for identifying SLP patterns of synoptic scale (i.e., locations of baric centers) and for identifying the humidity source regions, which are essential while analyzing the synoptic reasons of extreme precipitation and flood events.

High precipitation causing extreme floods is usually associated with cyclones and accompanying frontal zones. The main goal of this study was to follow the formation and relocation of these cyclones and thus, to determine the source regions of abundant PW supplies, which resulted in heavy rain in the Sudeten preluding flood incidents. To this end, standardized SLP anomalies for the 5 days preceding floods were used and the 17 flood cases were divided into five groups varying by the evolution of the SLP patterns before floods. The division was performed using one of the clustering techniques, namely the Ward's method [36,37]. Ward's minimum variance method (or simply Ward's method) is a popular hierarchical clustering method. As a hierarchical method it begins with *n* single-member groups, and merges two groups at each step, until all the data are in a single group after *n* − 1 steps. The criterion for choosing which pair of groups to merge at each step is minimizing the sum of squared distances between the points and the centroids of their respective groups, summed over the resulting groups. In effect, Ward's method minimizes the sum (over the *K* dimensions of an element *x*) of within-groups variances [37]. This method is most frequently used in climatic classifications [36,37] to identify the atmospheric circulation patterns associated with the occurrence of specific weather phenomena. The graphical results of the Ward method is a tree diagram, which illustrates every stage of hierarchical clustering. A practical problem appears to decide which stage of clustering should be chosen as the final solution. According to Wilks [37], however, the principal goal is to find the level of clustering that maximizes similarity within clusters and minimizes similarity between clusters; in practice the best number of clusters is usually not obvious. Wilks [37] postulates that determining the number of groups requires a subjective choice that depends on the goals of the analysis.

In the case of this study, the clustered objects were the flood events, the standardized daily SLP anomalies in the days preceding floods performed the variables and five almost equally numbered groups were distinguished (Figure A1). For each of the obtained groups composite SLP anomaly maps for the days preceding floods were constructed. Following the location of the SLP anomaly centers in consecutive days before the day of the flood culmination, the cyclone tracks were detected. Besides SLP anomalies, contour maps of PW anomalies were also drawn for the same days preceding floods. Fields of PW anomalies allowed indicating the source regions of water supplies which were involved in flood incidents.

Additionally, back trajectories of air masses for the selected cases were constructed, using the NOAA HYSPLIT model [38,39]. The model analyzed air masses movement for three altitudes above ground level: 500, 2000 and 5000 meters. The analysis of air trajectory at the three altitudes provided significant input to the information obtained from the composite low level pressure maps and made it possible to identify probable source area of air masses causing abundant precipitation in the Sudetes.

## 4. Results

### 4.1. Average Hydrometeorological Conditions in the Study Area

The mean annual precipitation total in the study area amounts to approximately 700 mm, and it increases with height above sea level to 1131 mm on Śnieżka. Precipitation in the summer half-year (May–October) is prevailing throughout the area, constituting 64% of annual precipitation. The spatial distribution of precipitation in the summer half-year is similar to the annual distribution of precipitation (Figure A2, left). The highest totals are recorded in the Karkonosze Mountains and Złote Mountains (>600 mm).

The annual river runoff ranges from less than 200 mm in the Sudeten foreland (the Nysa Szalona and Skora rivers) to almost 1000 mm on Sudeten mountains streams (the Kamienna River). In the majority of the catchments, runoff of the winter half-year (November–April) prevails, only in the Biała Lądecka and Łomnica river catchments, runoff of the summer half-year adds up to more than 50% of its annual value. Runoff of the summer half-year has a similar spatial distribution to the annual runoff. The highest values are recorded on rivers in two regions. In the Eastern Sudetes, values higher than 350 mm are characteristic of the Biała Lądecka and Wilczka rivers originating from the Złote Mountains and the Śnieżnik Massif. Even higher values of runoff in the summer half-year are recorded on the Karkonosze mountains streams—the Kamienna (450 mm) and Łomnica with Jedlica rivers (up to 400 mm) (Figure A2, right). The seasonal variations of water discharge are presented by the Pardé coefficient: $Q_m/Q_{yr}$, where $Q_m$ is mean monthly water discharge in $m^3s^{-1}$, and $Q_{yr}$ is mean annual water discharge in $m^3s^{-1}$. According to the typology of the Polish river regimes by Dynowska [40], most rivers of the Sudeten Mountains are characterized by the nival–pluvial regime, with the Parde's coefficients reaching 130–180% in the spring months, and exceeding 100% in the summer months [41]. In the period 1971–2015 the seasonal structure of runoff of the analyzed rivers reveals highest values of the Pardé coefficient in the spring season (March–April). However, there is also an increased runoff in summer resulting from rainfall (Figure 2). The summer high-water stages induced by precipitation are characterized by violent course and pose a large flood threat.

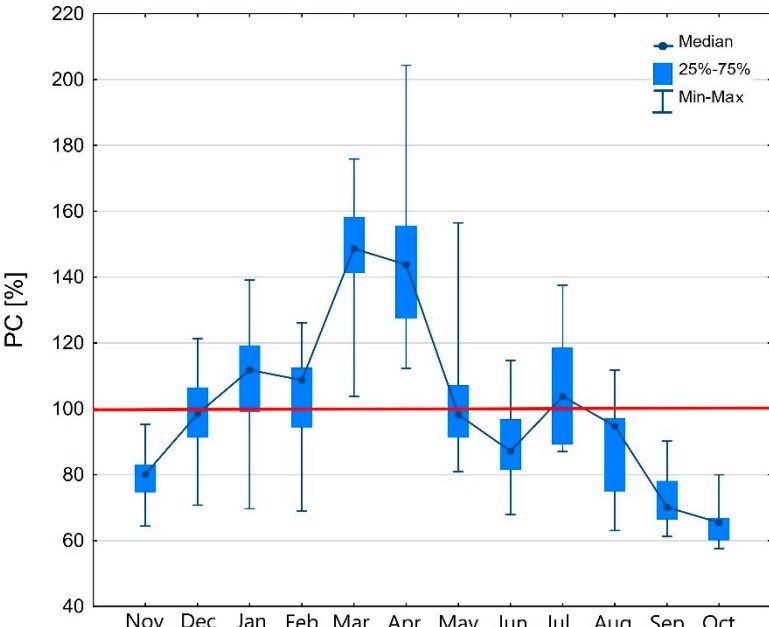

**Figure 2.** Range of variability of the Pardé's coefficients (PC) of the analyzed rivers in Polish Sudeten in 1971–2015. In the boxplot, monthly median values are marked with black dots, range of the 1st and 3rd quartile are marked by box, and the whiskers show extreme values.

### 4.2. Hydrometeorological Conditions during Flood Events

Based on methodology described in the previous section, a total of 17 cases of high-water events, which occurred in 14 summer seasons, were identified. In three seasons (1975, 1977 and 1997) synchronous floods on all the investigated rivers appeared. Some basic characteristic features of each flood case are given in Table A1.

The studied mountain rivers, due to the varied terrain, poor ground permeability and low retention capabilities of their catchments, react very quickly to water supply from precipitation during the summer months. In the analyzed cases, the maximum daily precipitation reached usually 50 to 100 mm in a flood-preceding week. On the mountain streams almost instant reaction was observed, and the culminations occurred usually after one to two days, like during the extreme flood in July 1997 (Figure 3). The increment of the flood discharge was usually quick and significant, from a few to even 50 $m^3$ $s^{-1}$ within 24 h in the Nysa Kłodzka River at gauge Międzylesie, and more than 250 $m^3s^{-1}$ in the Biała Lądecka River at gauge Lądek Zdrój, recorded during the largest flood in July 1997 (Figure 3). Almost synchronous occurrence of floods in the Nysa Kłodzka River and its tributaries in the Kłodzka Valley caused the overlap of the flood waves. This resulted in the appearance of the culmination wave with the maximum discharge with the probability of occurrence 0.1%.

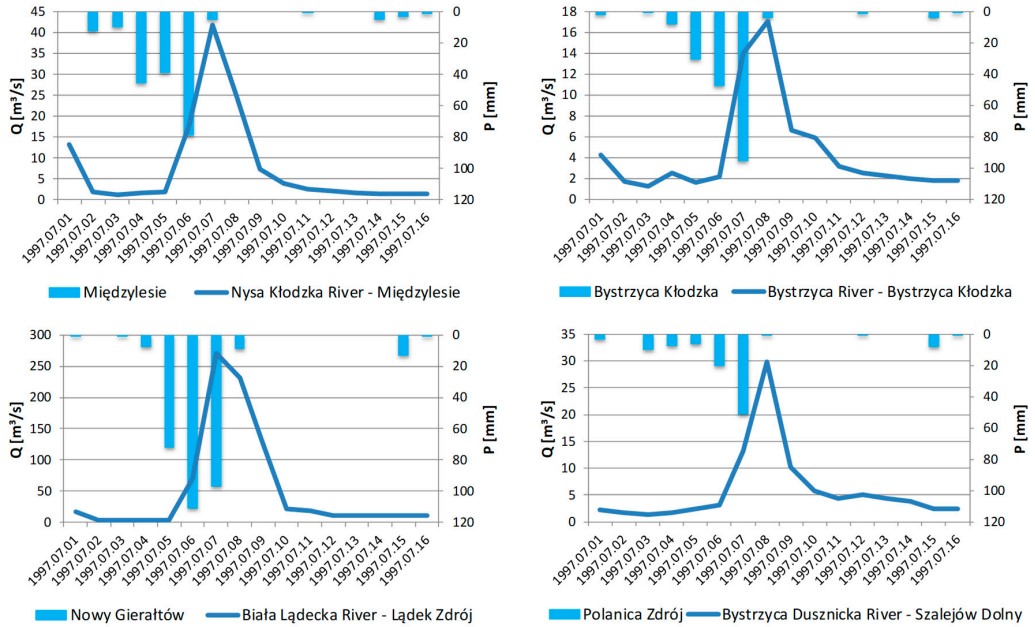

**Figure 3.** Course of daily discharge ($m^3s^{-1}$) (blue line, left axis scale) and daily precipitation totals (mm) (light blue bars, right axis scale) in exemplary locations during flood in the Kłodzko Valley in 1997.

Mean precipitation totals in the study area during the seven-day periods preceding the moment of flood culmination varied from 54.5 mm (28.05–3.06.2010) to 161.4 mm (2–8.08.2006) and in most cases the extreme daily precipitation (usually exceeding 50 mm) appeared one or two days before the discharge culmination (Figure 3). The maximum weekly precipitation total was 412.6 mm, recorded in 2006 in Sowie Mountains, on the precipitation station in the Walim village. High precipitation reaching above 300 mm was recorded during floods in 1977 (345.6 mm; 28.07–3.08) and 1997 (312.8 mm; 2–8.07 and 305.7 mm; 14–20.07). Regions of heavy precipitation include the area of Kamienne Mountains and Sowie Mountains, as well as Złote Mountains and Śnieżnik Massif (Figure A3).

### 4.3. Pressure Patterns and Precipitable Water Content over Euro-Atlantic Sector during Flood Events

In the Euro-Atlantic sector, SLP in the summer reaches the highest values over the Atlantic Ocean west of the Azores (>1023 hPa), and gradually decreases northwards. The center of the low-pressure system is located over the north Atlantic, southwest of Iceland (<1010 hPa). A high horizontal

baric gradient occurs between the aforementioned baric systems over the ocean, decreasing over the continent.

The average summer spatial distribution of PW in the Euro-Atlantic region reveals strong dependence on vicinity of source regions, i.e., warm sea/oceanic waters and on thermal conditions. Pursuant to the Clausius–Clapeyron formula, warmer air can contain more water vapour, the PW decreases in the northernmost part of the studied area and also with height a.s.l., which is related to lower temperature, as well as to lower thickness of the troposphere over mountainous regions (Figure 4).

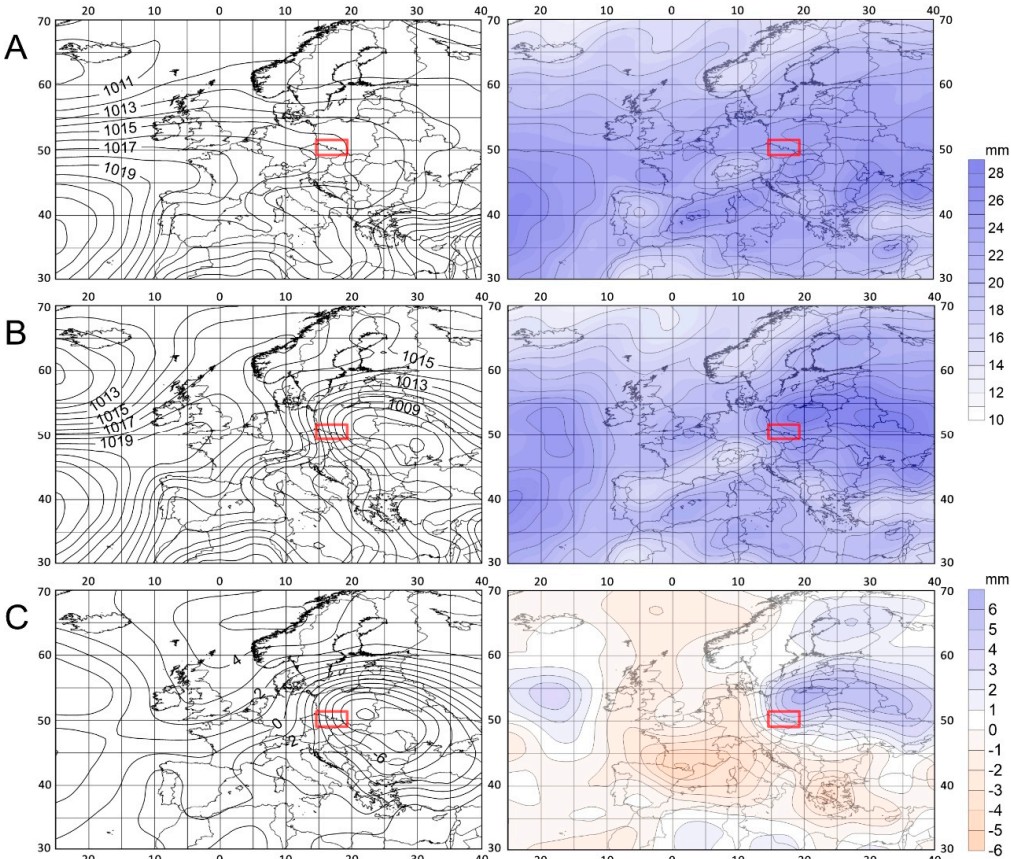

**Figure 4.** Mean see level pressure (SLP) (hPa) ((**A**), left) and precipitable water (PW) (mm) ((**A**), right) in the summer season, and mean SLP ((**B**, left) and PW ((**B**), right) on the flood culmination day and their anomalies (**C**); (**B**,**C**)—composites of all 17 cases of flood. Region of Polish Sudeten Mountains marked as red rectangles.

In the days of flood culmination, substantial departures from average SLP field are observed. Over the Atlantic Ocean, in the area of the Azores, a high pressure system persisted, with a wedge reaching north Europe. In the center of this anticyclone, SLP exceeded 1024 hPa. Over the northern areas of the Atlantic, a low-pressure system occurred with a center southwest of Iceland (<1010 hPa). Simultaneously, a major part of the central, eastern, and southern Europe was under the influence of a deep low-pressure system with a center over Ukraine (<1006 hPa). On the analyzed days over the Atlantic and a major part of north and north-west Europe, SLP was higher than average, maximally by more than 5 hPa over the Norwegian Sea (Figure 4). Nonetheless, negative anomalies were recorded over the majority of the area, with a center persisting over southeast Poland (<−8 hPa). Two evident areas of high PW values can be designated on the analyzed days, namely central-east Europe and the Atlantic Ocean. The highest anomalies were recorded over the former area. Over northeast Poland

they were >6 mm. The range of the anomalies extended from east Germany to west Russia. Moreover, higher than average PW also occurred over the northeastern regions of the continent and the ocean.

The above analysis of baric conditions concerned only the averaged situation for all analyzed cases and it emphasized occurrence of positive anomalies of PW and negative anomalies of SLP over eastern Europe, which evidenced occurrence of low-pressure systems. Further research showed that the occurrence of floods in southwest Poland was related to five different types of SLP patterns (Figure A1, Table A1). In all the types, on the flood culmination day over central Europe, low-pressure systems occurred, as suggested by lower than average SLP, but the cyclones had different location and extent, intensity, and they came by different cyclone tracks. In the majority of the analyzed types, the range of the center of SLP anomalies covered the southeastern or eastern regions of Poland. Over the ocean or northern regions of the continent, centers of positive anomalies persisted, suggesting the presence of high pressure systems forcing the movement and location of lows (Figures 5 and 6).

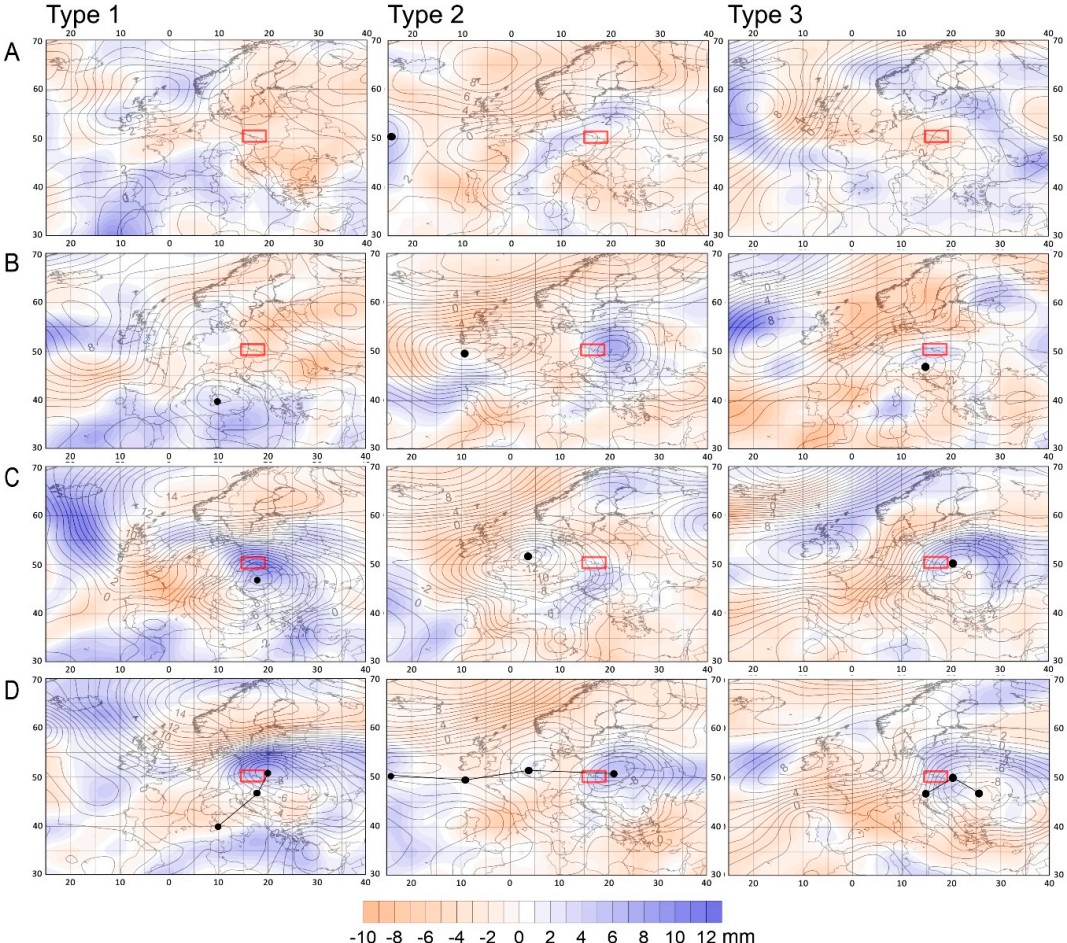

**Figure 5.** Circulation types 1, 2 and 3 mean SLP (hPa) (solid lines) and PW (mm) (colour scale) anomalies 6 days (**A**), 4 days (**B**), and 2 days (**C**) before the flood culmination and on the flood culmination day (**D**); black dots indicate centers of negative SLP anomalies and in column D cyclone tracks are marked; region of Polish Sudeten Mountains marked as red rectangle.

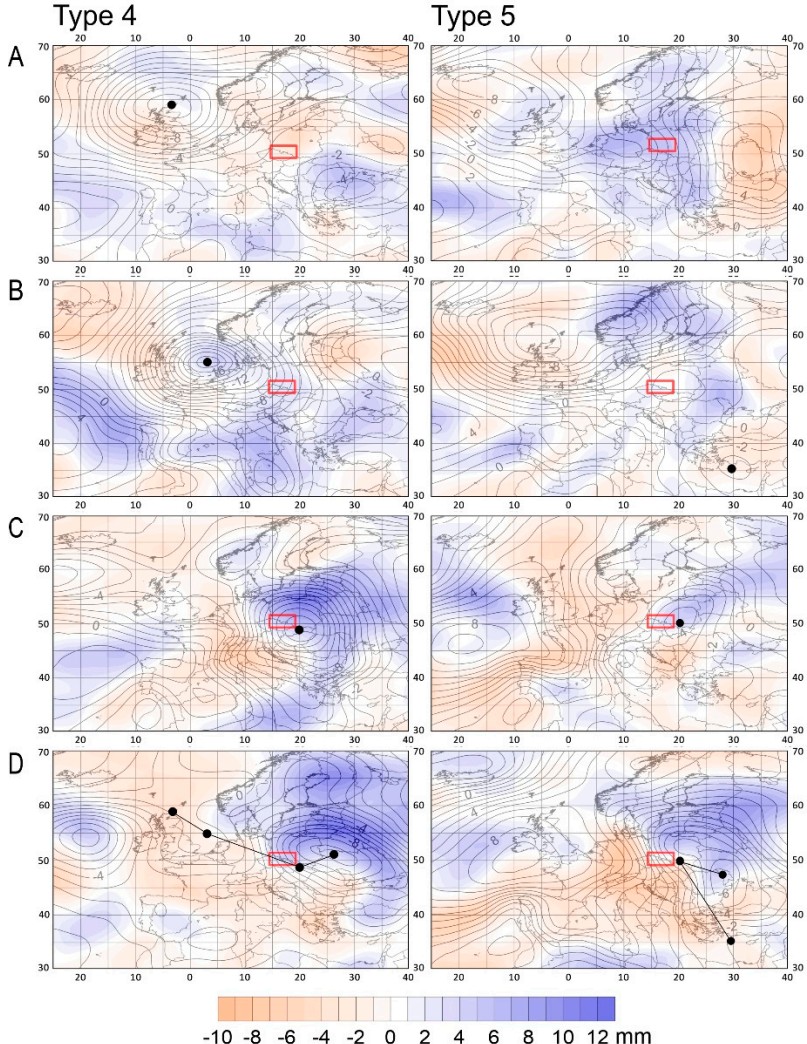

**Figure 6.** Circulation types 4 and 5 mean SLP (hPa) (solid lines) and PW (mm) (colour scale) anomalies 6 days (**A**), 4 days (**B**), and 2 days (**C**) before the flood culmination and on the flood culmination day (**D**); black dots indicate centers of negative SLP anomalies and in column D cyclone tracks are marked; region of Polish Sudeten Mountains marked as red rectangle.

In type 1, consisting of three flood cases (Figure A1), the center of negative SLP anomalies was located over southeast Poland (<−8 hPa). A low-pressure system was detectable also in the middle troposphere, where the center of negative Z500 anomalies (<−80 m) was located south to the low-level cyclonic center (Figure 7). A major part of the continent was simultaneously within the range of a high, as suggested by positive SLP and Z500 anomalies with a center over Scandinavia (>17 hPa). Such a distribution of baric systems caused blocking of a low moving from over the Mediterranean Sea. An evident low-pressure system with a center over the Tyrrhenian Sea (<−3 hPa) was recorded approximately four days before the occurrence of the flood culmination in the Sudetes. The moving cyclonic system strengthened, as suggested by growing anomalies (both SLP and Z500) which two days before the analyzed days increased to <−10 hPa (center over Slovakia and Hungary) and −140 gpm (center over the northern Italian Peninsula). The described baric situation generated advection of humid air masses from the south and southeast, as suggested by the location of positive PW anomalies. The center of the PW anomalies (>9 mm) two days before the flood culmination in the Sudetes was located over the analyzed area. On the following days, they moved further north, and on the day of culmination, the center was located over north Poland (>9 mm).

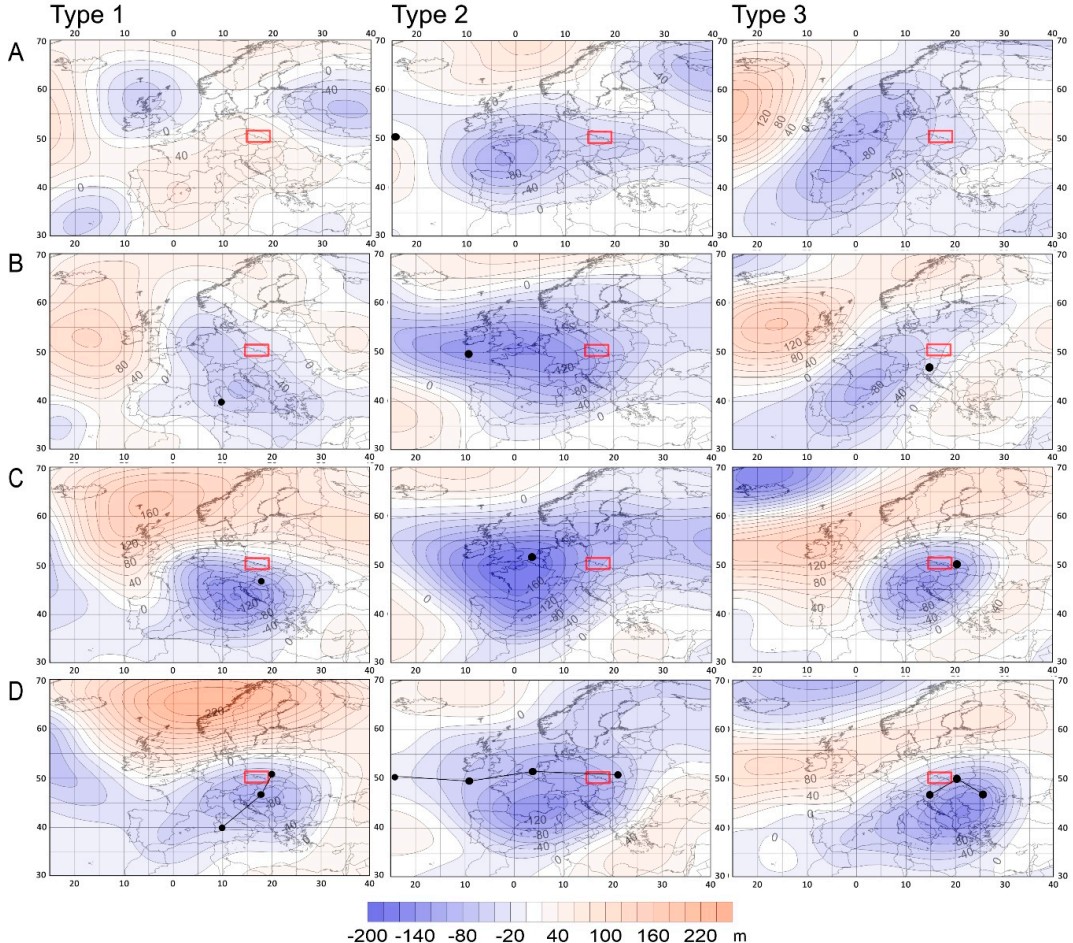

**Figure 7.** Circulation types 1, 2, 3 mean Z500 anomalies (m) 6 days (**A**), 4 days (**B**), and 2 days (**C**) before the flood culmination and on the flood culmination day (**D**); black dots indicate centers of negative SLP anomalies (repeated from Figure 5) and in column D cyclone tracks are marked; region of Polish Sudeten Mountains marked as red rectangle.

A different baric situation was recorded in type 2, consisting of three flood cases (Figure A1) during which a major part of the continent was within the range of negative SLP anomalies, with the center, similarly as in type 1, over southeast Poland (<−9 hPa) (Figure 5). The center of Z500 anomalies (<−120 m) was located southwest to SLP anomalies, i.e., over southern France, triggering southwestern airflow in the Polish Sudeten at the middle-tropospheric level over the Polish Sudeten (Figure 7). An area of higher than average pressure simultaneously occurred over north Atlantic on the analyzed days. In this type, a low-pressure system developed and then moved from over the Atlantic Ocean. The commencement of its development was observed approximately six days before the occurrence of the flood culmination in the Sudetes. The moving low-pressure system strengthened, and two days before the flood culmination in its center SLP was lower than average by more than 13 hPa and Z500 was located lower-than-average by 160 m. The occurrence of the described low was preceded by the occurrence of a weaker low-pressure system extending over central Europe. The moving low caused advection of humid air masses from over the Atlantic Ocean. The direction of advection is indicated by the course of movement of positive PW anomalies. The highest anomalies were recorded on the flood culmination day, and their maximum values particularly covered east Poland (>5 mm).

In type 3 (four flood cases, Figure A1), the Euro-Atlantic sector was under the influence of an anticyclonal center with positive SLP and Z500 anomalies in the west and cyclonic center with negative SLP and Z500 anomalies in the east (Figures 5 and 7. The center of positive SLP anomalies persisted over Ireland (>8 hPa) and Z500 (100 m) west to it, and the center of negative anomalies was located southeast of Poland (−8 hPa, −140 m). The beginning of development of a low-pressure system, occurring in central Europe, was observed over the boundary of the Italian and Balkan Peninsula four days before the occurrence of the flood culmination in the analyzed area. Then, the cyclone moved to the northeast, and two days before the flood culmination, the center of negative SLP anomalies persisted over southeast Poland (<−7 hPa). The direction of this short cyclone track was forced by the blockage situation over the western and northern regions of the continent. Such a baric situation generated advection of humid air masses from the south. Like in the case of SLP anomalies, the highest PW anomalies were recorded two days before the occurrence of the flood culmination, and their center persisted over Belarus (>7 mm). On the following days, the anomalies moved further west, and on the day of culmination, their center was located over north Poland (>4 mm).

In the next type of circulation, namely type 4, consisting of four flood cases (Figure A1), two low-pressure systems were present in the Euro-Atlantic sector, separated by an area of higher than average pressure. The first center of negative anomalies persisted over central-east Europe with a center over the border of Poland, Belarus, and Ukraine (<−10 hPa), and the other, somewhat weaker and not important for the studied matter, was located over the ocean (<−8 hPa) (Figure 6), both of them detectable also in the middle tropospheric level (Figure 8). In this type, similarly as in type 2, the development of a low-pressure system was observed over the Atlantic Ocean, although the initial area was shifted north compared with type 2. Then, the system moved to the southeast. In its center, the highest SLP and Z500 anomalies were recorded four days before the occurrence of the flood culmination in the Sudetes (<−18 hPa and <−160 m). Over the following days, the moving anomaly field weakened. The distribution of PW anomalies suggests advection of humid air masses, initially from the southwestern sector, and then from the southern sector. The highest PW anomalies over the analyzed area were recorded two days before the occurrence of the flood culmination. Their maximum values were recorded over north Poland and Russia (>9 mm). On the following days, the resources of precipitable water moved further to the east (Figures 6 and 8).

In type 5, consisting of three flood cases (Figure A1) like in the types described above, two baric systems were dominant in the Euro-Atlantic sector on the day of flood culmination. The prevailing area of central, south, and east Europe remained within the range of negative SLP/Z500 anomalies with a SLP anomaly center extending from Ukraine to Bulgaria (<−6 hPa) and Z500 anomaly center extending from Hungary to Adriatic Sea (<−120 hPa). Over the remaining area, air pressure was higher than average. The beginning of development of the low was observed over the eastern regions of the Mediterranean Sea Basin approximately four days before the occurrence of the flood culmination in southwest Poland. On the following days, the low-pressure system moved to the north-west and was slightly strengthened. The occurring baric situation forced advection of humid air masses from the south. The additional presence of the high over the ocean and west Europe forced advection of air masses from over the Atlantic. The highest PW anomalies over Polish Sudeten were recorded approximately two days before the flood culmination (Figures 6 and 8).

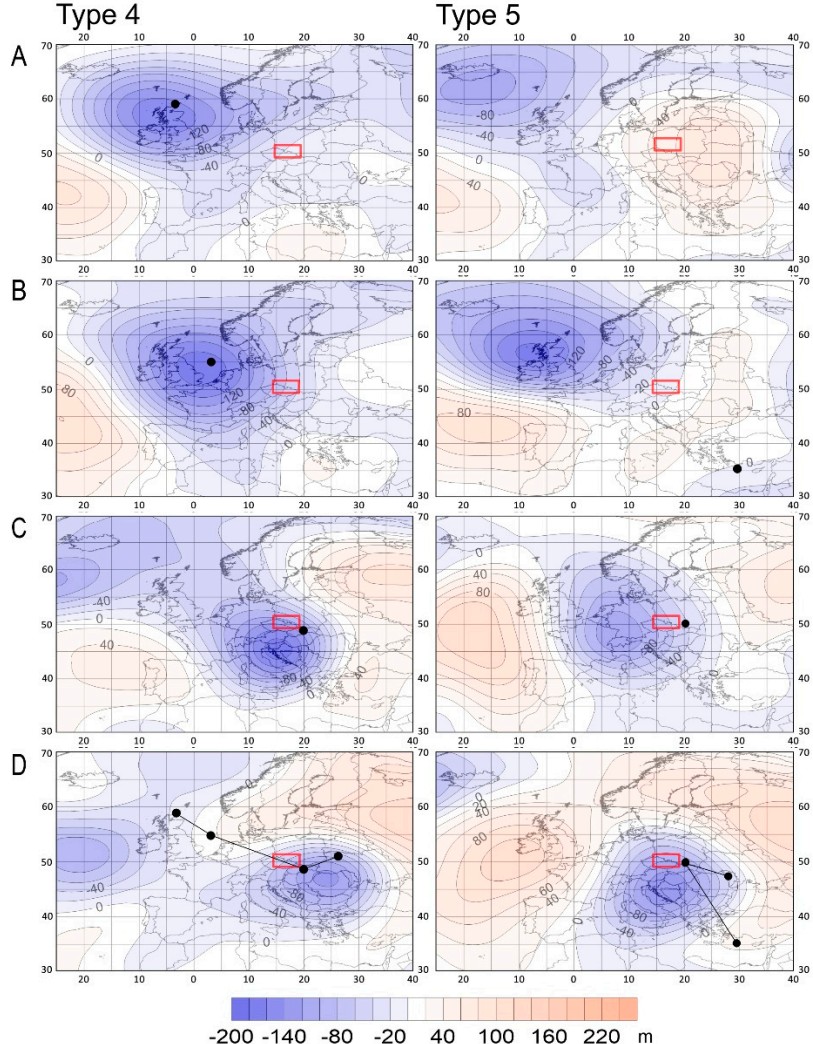

**Figure 8.** Circulation types 4 and 5 mean Z500 anomalies (m) 6 days (**A**), 4 days (**B**), and 2 days (**C**) before the flood culmination and on the flood culmination day (**D**); black dots indicate centers of negative SLP anomalies (repeated from Figure 6) and in column D cyclone tracks are marked; region of Polish Sudeten Mountains marked as red rectangle.

### 4.4. Trajectories of Air Masses before Flood Events

The simultaneous occurrence of very high water stage on many rivers in southwest Poland was related to the presence of low-pressure systems over central-east Europe, causing advection of humid air masses. The prepared maps were supplemented by 72-hour backward trajectories of air particles for the day of the flood culmination (Figure 9). In the majority of flood cases, (besides the ones belonging to the type 2), during three days preceding flood culmination, the inflow of air particles from the northern sector was predominant in the lower layers of the troposphere. The northern inflow, almost regardless of the represented type is a consequence of the counter-clockwise circulation around cyclonic centers. In higher layers of the troposphere, advection from the south was predominant, excluding cases belonging to type 2 and 3. During the flood from July 1980, when advection from over the Atlantic Ocean was observed throughout the tropospheric profile. The aforementioned case was qualified to the designated type 2, in which a pronounced western cyclone track was recognized. Also the case from July 1997, belonging to the type 3, revealed a different circulation of air particles, i.e., the inflow from the northern sector. In all of the analyzed cases, rising of air masses was recorded, which is typical of cyclonic systems and atmospheric fronts (Figure 9).

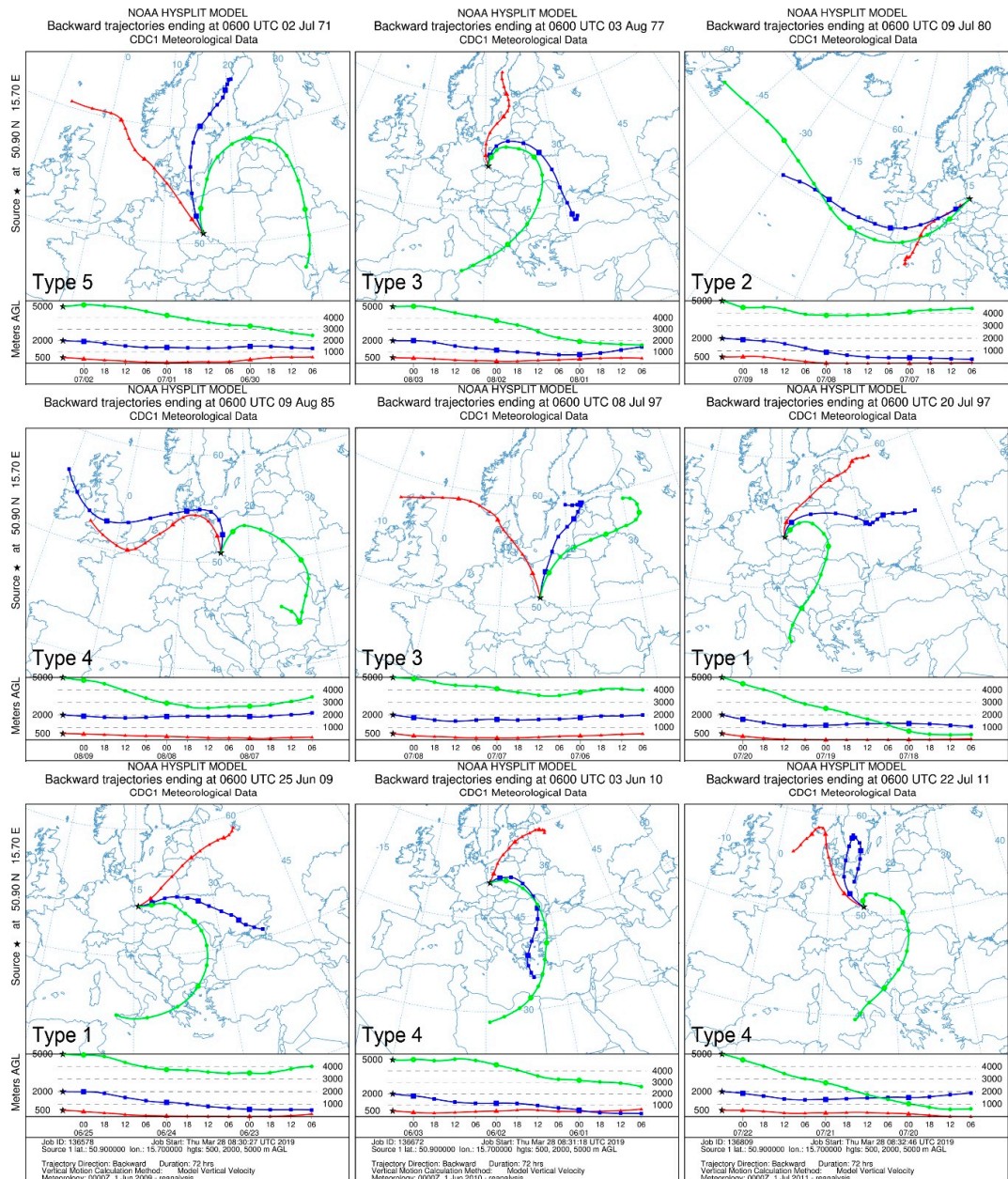

**Figure 9.** Seventy-two-hour backward trajectories of air parcels determined for the flood culmination day with the circulation type indicated.

## 5. Discussion

The Sudeten Mountains and its foreland are considered one of the most vulnerable to floods regions in Poland, mainly due to physiographical features such as various landforms and geology [42,43]. However, meteorological conditions are the most important factors that govern time of occurrence and magnitude of floods in this area (confirmed also in previous studies [23–30]). In each case of extremely high runoff in the majority of the Sudeten rivers, the maximum recorded runoff followed immediately (usually with an about one-day delay) extremely high precipitation in this area. A rapid hydrological reaction to the meteorological factor is explained by low retention capabilities of the mountain catchments due to the landform attributes (steep slopes and narrow river valleys) and poor ground permeability.

Despite fast reactions to precipitation in mountain catchments, the key issue in studying atmospheric forcing of floods in the Sudetes is investigating synoptic conditions which lead to

abundant rainfall preceding each flood event. Slow-evolving synoptic conditions with the strong cyclonic flow induced by the deep lows were recognized as the most important factors for strong, flood-bringing precipitation in various mountain regions [20,21]. Apparently, in the Sudeten, slow evolving low-pressure systems are responsible for heavy precipitation, but they could be of different origin, various extent and intensity. The anomaly-based analysis applied in this study allowed following the development, location and movement of the cyclonic centers in the days preceding floods. The anomaly-based approach has been employed previously in research concerning atmospheric forcing of extreme events [44–47].

Five types of cyclonic circulation patterns responsible for extreme runoff in the Sudetes were identified in this study. Three of them commence with development of Mediterranean cyclones, which thereafter move northwards providing large amounts of precipitable water in central Europe. According to Degirmendžić and Kożuchowski [48] precipitation associated with Mediterranean cyclones accounts for about 10% of the total amount of precipitation in Poland, and this share increases for the extreme daily precipitation totals. Cyclones of southern origin bring exceptionally abundant rainfall, especially in the warm season, and they are responsible for the highest daily precipitation totals recorded in Poland [48]. Niedźwiedź et al. [17] identified three cyclonic circulation types triggering the most extreme rainfall in the Tatra Mountains, all of them connected to stationary cyclones originating in the Adriatic Sea and propagating with van Bebber's track Vb [49]. The role of cyclone pathway "Zugstrasse Vb" in triggering extreme precipitation and floods was confirmed also for other regions in Europe [2,50]. It has been proven in this study that beside Mediterranean lows, also cyclones of western origin may develop flood-activating rainfall in the Sudetes; however, due to the counterclockwise circulation within these low-pressure systems, the Mediterranean is the source region of humidity transported northwards, even if the cyclone itself moves along the zonal western path.

Investigating the distribution of precipitable water anomalies, accompanying and preceding analyzed events, allowed recognizing a strong meridional northward transport of moisture, which took place in most cases of abundant rainfall, followed by floods on the Sudeten rivers. This can be compared to so-called atmospheric rivers—a phenomenon well recognize in the climatological and hydrological literature—which are defined as narrow corridors responsible for the majority of the poleward water vapor transport across the midlatitudes [51–56]. However, they differ from atmospheric rivers by the non-filament character and lower intensity. Atmospheric rivers form a part of the broader warm conveyor belt of extratropical cyclones and they are characterized by high water vapor content. According to Lavers and Villarini [52,53] atmospheric rivers do not affect precipitation in southern Poland in summer. The landfalling atmospheric rivers have the most impact on western Europe. However, they reach as far east as central Europe (Germany, Poland—mainly the northern part) in late spring, winter and autumn. In summer their influence on European precipitation extremes is diminished [56]. Bosart et al. [57] and Moore et al. [58] postulated that extreme weather events including abundant precipitation in North America, should be linked to high-amplitude upper-level flow patterns. A rapid transition from a strong zonal flow configuration to a meridional flow configuration and then the Rossby wave dispersion and breaking (often manifested by specific formation of the potential vorticity streamer) establish favorable conditions for extreme weather events in North America [57,58].

The intense meridional water vapor transport is, on one hand, crucial for water resources, but on the other hand it can also cause disastrous floods, especially when encountering mountainous terrain, like in cases described in this study. The mesoscale mechanisms for orographic precipitation enhancement in association with the passage of cyclones and frontal systems were recognized by Medina et al. [18]. They described a shared layer, which forms at the intermediate sub-crest heights, as an essential component of the orographic precipitation.

For most of floods events in the Polish Sudetes a particular synoptic pattern of air circulation was documented, namely northern and/or northeastern advection of cool air masses in the low troposphere and southern advection of warm and humid air masses in the middle troposphere. The humid air masses of southern origin lift slowly over the colder air-layer while moving northwards, which indicates

a typical warm front structure. The relevance of the upper tropospheric transport was emphasized by Wypych et al. [22] who analyzed synoptic determinants of extreme precipitation events in the Polish Carpathians. They found a similar pattern of air flow, causing extreme rainfall, as recognized in this study, namely, the collision of warm southerly air which transports huge amounts of water in the mid-troposphere, with cool air from the north. Such pattern reminds to a certain degree, the conditions of so-called Spanish Plume over the United Kingdom, which appears in the summer months and leads to extreme weather events such as extreme high temperatures, intense rainfall with potential for flash flooding and damaging hail storms, etc. [22,59,60].

## 6. Conclusions

Atmospheric processes and mechanisms that lead to extensive floods in the Polish Sudetes were recognized in this study. The results confirmed a direct impact of heavy precipitation on extremely high runoff episodes, and an essential role of synoptic conditions in triggering abundant rainfall. Cyclonic circulation patterns of different origin, various extent and intensity, are responsible for heavy precipitation, which induces floods. Five typical storm tracks were recognized, starting several (four to six) days before flood culmination. The anomaly-based method was used to investigate the evolution of the pressure field in low and middle troposphere and the moisture content. Maps of anomalies, enabled recognizing an early formation of the rain-bringing cyclones and allowed distinguishing areas of positive departures of precipitable water content. A particular pattern of air circulation, involving different tropospheric levels, was recognized in most of analyzed flood cases. Namely, the warm humid air transported from Mediterranean in the middle troposphere overlapped with the cool air masses transported from north/northeast in the low troposphere, forming a typical frontal structure. Recognizing the specific meteorological flood-triggering mechanisms, involving evolution of pressure patterns, changing in atmospheric moisture and occurrence of precipitation, may contribute to better understanding atmospheric forcing of floods in mountain areas. An early detection of negative centers of sea level pressure could give a chance to predict heavy rain episodes and following flood events.

Future research directions motivated by the current study should concentrate on investigations of whether the same cyclone tracks or similar to those distinguished in this study establish favorable conditions for extreme and flood-threating precipitation in other regions of central Europe. A closer investigation of integrated water vapor transport within the mid-latitude cyclones would allow to validate the role of atmospheric rivers in meridional moisture transport in central Europe. Additionally, a broader set of factors that contribute to occurrence of extreme weather events could be taken into consideration, for example, upper-level weather patterns, including jet stream and Rossby waves modifications [57,61–63].

**Author Contributions:** Conceptualization, E.B., D.W., A.M.T. and D.J.; methodology, E.B., D.W., A.M.T. and D.J.; validation, E.B., D.W. and A.M.T.; formal analysis, D.W. and A.M.T.; writing—original draft preparation, E.B., D.W. and A.M.T.; writing—review and editing, E.B., D.W. and A.M.T.; visualization, D.W. and A.M.T.; supervision, E.B.

**Funding:** This research received no external funding.

**Conflicts of Interest:** The authors declare no conflict of interest.

## Appendix A

**Table A1.** Floods characteristics with the type they belong to, beginning and end dates, maximum precipitation in the week preceding flood culmination (with a name of station).

| Type | Beginning | End | Max Precipitation (mm) | Name of Station |
|------|-----------|-----|------------------------|-----------------|
| 5 | 26.06.1971 | 02.07.1971 | 170.7 | Ciechanowice |
| 3 | 26.06.1975 | 02.07.1975 | 141.2 | Śnieżka |
| 3 | 28.07.1977 | 03.08.1977 | 345.6 | Śnieżka |
| 2 | 17.08.1977 | 23.08.1977 | 138.9 | Jarnołtówek |
| 2 | 03.07.1980 | 09.07.1980 | 199.1 | Nowy Gierałtów |

**Table A1.** *Cont.*

| Type | Beginning | End | Max Precipitation (mm) | Name of Station |
|---|---|---|---|---|
| 5 | 14.07.1981 | 20.07.1981 | 266.4 | Rębiszów |
| 4 | 03.08.1985 | 09.08.1985 | 174.3 | Nowy Gierałtów |
| 3 | 02.07.1997 | 08.07.1997 | 312.8 | Jarnołtówek |
| 1 | 14.07.1997 | 20.07.1997 | 305.7 | Boguszów-Gorce |
| 2 | 15.07.2001 | 21.07.2001 | 172.4 | Przesieka |
| 4 | 08.08.2002 | 14.08.2002 | 179.5 | Ciechanowice |
| 3 | 02.08.2006 | 08.08.2006 | 412.6 | Walim |
| 1 | 19.06.2009 | 25.06.2009 | 211.5 | Nowy Gierałtów |
| 4 | 28.05.2010 | 03.06.2010 | 98.4 | Nowy Gierałtów |
| 1 | 22.09.2010 | 28.09.2010 | 126.0 | Bierna |
| 4 | 16.07.2011 | 22.07.2011 | 156.1 | Stara Kamienica |
| 5 | 20.06.2013 | 26.06.2013 | 122.3 | Stara Kamienica |

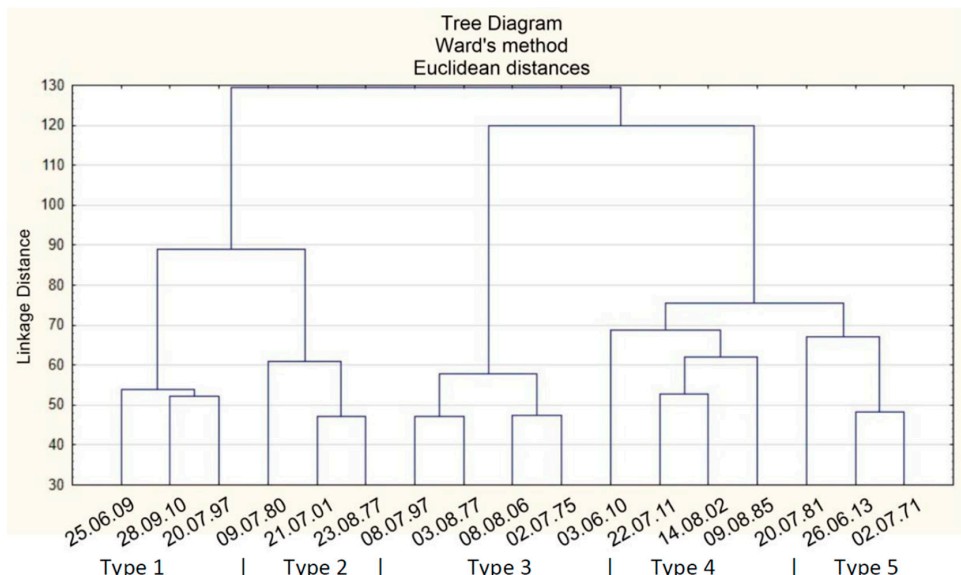

**Figure A1.** Graphical illustration of Ward's method—the tree diagram with the assigned types (dates indicate culmination days of floods belonging to each group).

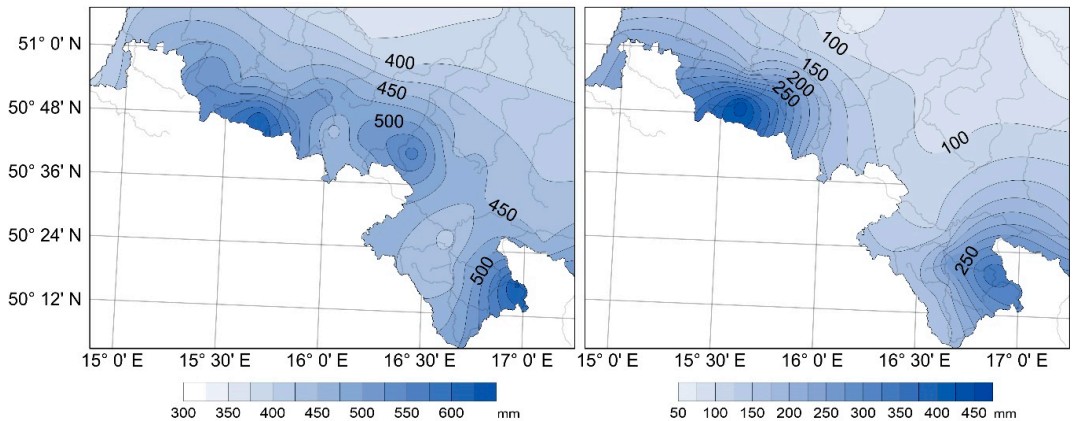

**Figure A2.** Mean summer half-year (May–October) precipitation (left) and runoff (right) totals.

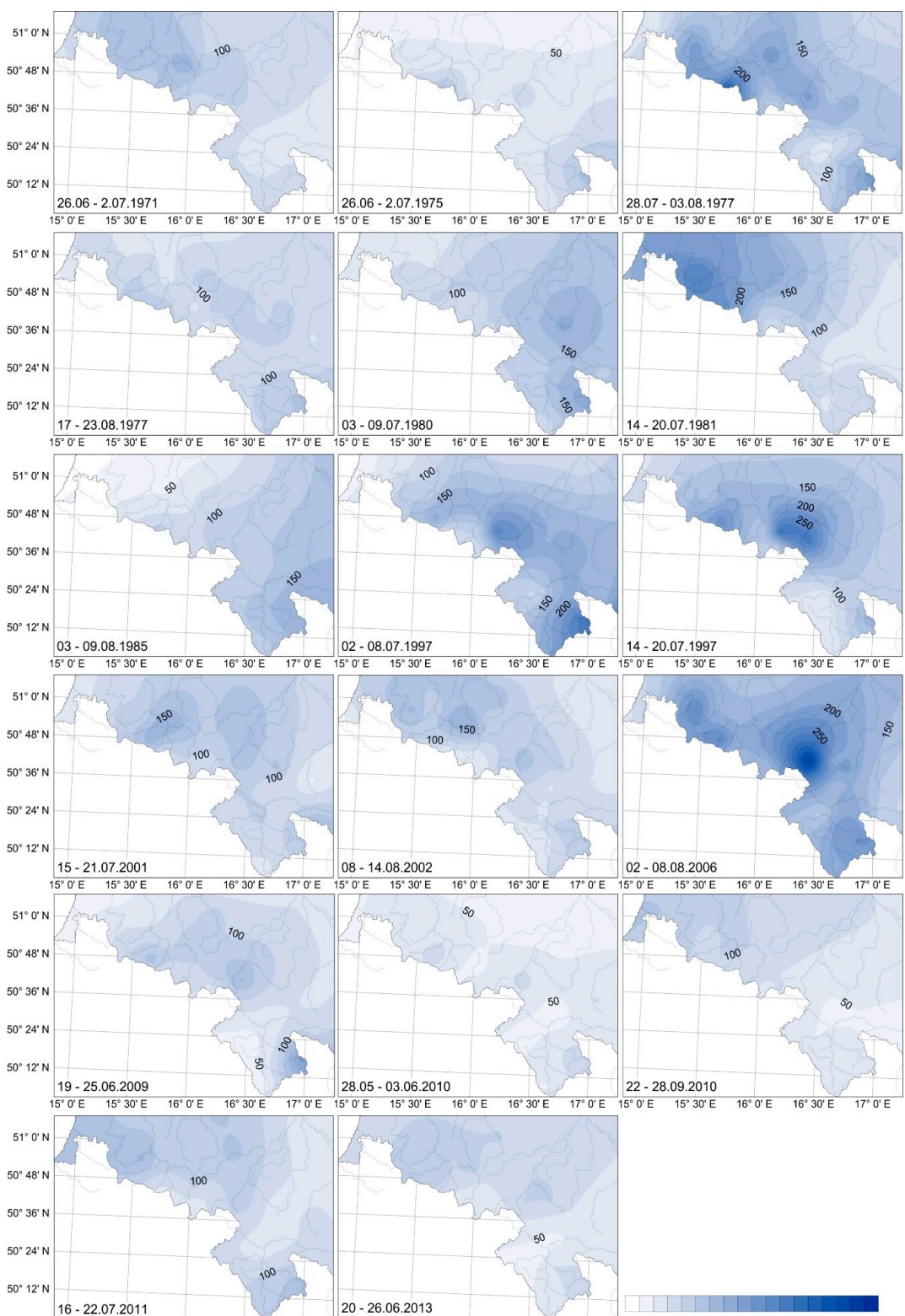

**Figure A3.** Precipitation totals during the 7-day periods preceding the day of flood culmination for each analyzed flood event.

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
