# Peer review of "Classification of Synoptic Conditions of Summer Floods in Polish Sudeten Mountains"

_water, doi:10.3390/w11071450_

Round 1
Reviewer 1 Report
Please find my comments in the attached document.

Author Response
We thank very much the Reviewer for taking time and effort in reading and analyzing our manuscript entitled “Classification of synoptic conditions of Atmospheric forcing on summer floods in Polish Sudeten Mountains” (previous title “Atmospheric forcing on summer floods in Polish Sudeten Mountains”). We are very grateful for all of the valuable comments, both general and detailed ones, which allowed to improve the manuscript. All of comments and remarks were considered carefully; the general comments allowed to revise our ideas concerning the studied issue and they are essential for our future studies.
We provided a record of changes introduced to the manuscript, according to all Reviewers’ suggestions. Changes are marked in the annotated version of the manuscript enclosed as a supplementary material (.pdf).
Attached below, please find our point-by-point responses to all of your comments.
Sincerely,
Ewa Bednorz and Co-Authors
General comments
Although the brevity of the manuscript is generally commended, there are important details that must be added in an updated version of the manuscript to provide context and insight to some of the important findings. Specifically, there must be more information provided regarding (A) the clustering technique [Ward’s Method] and (B) the upper-tropospheric weather composite of each of the clustered groups.
· Regarding issue (A), it is important for the reader to know how Ward’s Method resulted in five categories using 17 flood cases (why not three or six types?). How does the reader know that overlap between the five categories has been minimized? In other words, how does this technique guarantee the maximized independence between the five categories? Inspection of Figure 6 appears as though Type 1 is related to Types 3 and 5 and that Types 2 and 4 are related. How many flood cases were binned in each category type? How significant are the types using Ward’s Method when, on average, each type has a total of only three or four flood cases? What method was used to determine the Type (or category) of a particular storm? What was the sorting mechanism?
Response: A broader description Ward method was introduced in the manuscript. Figure A1, showing results of hierarchical clustering by the Ward’s method, with the composition of each group, was added in the Appendix (Fig. A1) and commented in the text. The Ward method was described in details in section 3. Considering the number of clusters, the principal goal is to find that level of clustering that maximizes similarity within clusters and minimizes similarity between clusters, however in practice the best number of clusters is usually not obvious. Wilks (2011) postulates that determining the number of groups requires a subjective choice that depends on the goals of the analysis. We intended to obtain not too big and possibly equally numbered groups and not too many of them, considering that we have 17 object to classify. As can be seen in the Figure A2 a larger number of groups was rather not possible. Establishing 4 groups would impel to join type 4 and 5 which differ significantly, the same problem would arise when dividing into 3 groups.
Reference: Wilks, D. Statistical methods in the atmospheric sciences. Elsevier Academic Press, 2011, Amsterdam
· Regarding issue (B), specifying the composite upper-tropospheric (e.g., potential vorticity [PV] on the 320 K isentropic surface) weather pattern for each of the clustered types may provide a context for the reasonableness of the Ward’s Method sorting and add to the general scientific literature on upper-level weather patterns giving rise to Extreme Weather Events (EWEs, Bosart et al. 2017) and/or Extreme Precipitation Events (EPEs, Moore et al. 2016). Both Bosart et al. (2017) and Moore et al. (2016) identified Rossby wave breaking (RWB), visualized as the curvature of the PV streamer on an isentropic surface, as an important precursor to the breakdown of the large-scale zonal to meridional flow pattern which provides the needed trajectories to bring anomalously humid air masses into the extra-tropics. Moore et al. (2016) also noted that large-scale weather patterns having anticyclonically curved PV streamers were more often precursors to heavy precipitation (EPEs) than patterns having cyclonically curved PV streamers. Of interest will be to re-examine the clusters (composites) of heavy precipitation events affecting the Polish Sudeten Mountains in the context of the upper-troposphere. Are each of the five types distinct in their associated upper-tropospheric PV streamers (trough/ridge tilt)? Does one type correspond with anticyclonic RWB, while another type corresponds with cyclonic RWB? Examination of SLP and SLP anomalies alone gives an incomplete picture as to the evolution of the relevant weather features during the 5-7 day period preceding the period of heavy precipitation. Examining upper-level weather maps will also provide a more complete context of the trajectory results displayed in Figure 7.
Response: In order to supplement our analysis with the upper-level weather patterns we added maps of anomalies of 500 hPa geopotential level (Fig. 6a,b). Composite anomaly maps constructed for each type picture meridional flow configuration during extreme precipitation events, as postulated by Bosart et al. (2017). We studied carefully works of Bosart et al. (2017) and Moore et al. (2016) and we found very interesting the upper-level flow patterns and mechanisms that establish favorable conditions for extreme weather events in North America, namely rapid transition from a strong zonal flow configuration to a meridional flow configuration and then the Rossby wave dispersion and breaking, manifested by specific formation of the potential vorticity streamer. However, we rather think that analyzing the Rossby waves and PV as influencing extreme weather events in Central Europe requires a new study; the concepts of Bosart et al. (2017) and Moore et al. (2016) will be essential and valuable for our future research.
Minor issues – editorial and technical issues
[1] (line 12) should read “…in the Polish Sudeten…”
[2] (line 21) should read “…form over the Mediterranean Sea and…”
[3] (lines 23-24) should read “…of the analysed cases…flood-triggering mechanisms involving…”
[4] (line 25) should read “…contribute to a better understanding of the atmospheric forcing…”
[5] (line 27, 62, 96 and elsewhere in the manuscript) “Polish Sudeten Mountains” (‘Mountains’ should be capitalized when one is referring to the name of a specific geographic feature).
[6] (lines 32-33) should read “Although Mudelsee et al. [1, 2] reveals that observations from Europe do not show…”
[7] (lines 35-36) should read “…Despite improved early warning…control infrastructure, flood damage has grown…flood-vulnerable areas have gained…”
[8] (line 38) should read “…there is widespread…but also flood hazards could be…”
[9] (line 39) should read “…justified by most of the model-based…”
[10] (lines 40-41) should read “…higher content of water vapor…”
[11] (line 43) should read “In contrast to droughts, which result from…”
[12] (lines 43-45) Floods are not necessarily a ‘short-duration phenomena’ (see your comment on line 87).
[13] (line 49) should read “…achieved through distinct…”
[14] (line 54) should read “…flood-producing synoptic patterns and may…”
[15] (line 56) should read “…floods in the southern, mountainous…”
[16] (lines 59-60) should read “…in July 1970 [21], July 1997 [22, 23] or May-June 2010 [23, 24].”
Response [1-16]: Appropriate corrections were made.
[17] (line 64) Please clarify the meaning of the term ‘concentration time.’
Response: We meant water concentration in river catchment, however the awkward term was deleted.
[18] (line 65) should read “This enabled the storm track determination of rain-brining cyclones,…”
[19] (line 67) should read “…flood-triggering mechanisms,…”
[20] (line 71) should read “…water resource regions…”
[21] (line 76) should read “…Klodzka Valleys…”
[22] (lines 77-78) should read “The slope of riverbeds in…exceeds 5% and, locally in the highest sections, reaches 1% or more [25].”
[23] (lines 82-83) should read “…the most violent and dangerous are summer floods,…”
Response [18-23]: Appropriate corrections were made.
[24] (lines 101-103) A brief description of the quality control that goes into the gauge observations of the IMGW-PIB databases would be helpful for the reader.
Response: All IMGW-PIB data undergo the quality control. As far as we know IMGW-PIB use control programs: WIRTHUAL\PROG\wirthual.exe, KLIM2002, and KLIMATY. Besides automatic detection of doubtful data, manual detection is proceeded, particularly on precipitation data. A brief comment on this issue was added in the appropriate section of the manuscript.
[25] (lines 104-106) Unusually brief (single-sentence) paragraphs should be avoided.
[26] (line 105) should read “…maps of weekly totals…”
Response [25-26]: Appropriate changes were made.
[27] (lines 105 and 135) Statements are inconsistent; seven days or five days preceding floods?
Response: These are just different matters. Precipitation totals were computed for 7 days preceding flood culmination (as a matter of fact, usually 1-3 first days of the week with small amount of rain, please see examples in Fig. 3). In the Ward’s method standardized SLP anomalies for the 5 days were employed and it was enough to find out the different cyclone tracks.
[28] (lines 107-114) Approach description is confusing.
Response: It was reformulated.
[29] (line 117) should read “…from the National Centre for…”
Response: Corrected.
[30] (lines 119-125) Why not include an equation showing how precipitable water is calculated? Also, the term ‘complex’ implies that the number is not a real number.
Response: Description of precipitable water was reformulated (using the definition from Glossary of American Meteorological Society) and equation added, according to suggestions of two reviewers.
[31] (line 129) A 2.5x2.5o grid resolves synoptic weather features, but does NOT resolve mesoscale weather features.
[32] (lines 132-133) should read “…and accompanying frontal zones.”
Response [31-32]: Appropriate changes were made.
[33] (lines 132-141) Please find my major concern item (A) described above.
Response: The Ward method was described in details in section 3. Please, see detail explanations above.
[34] (lines 144-146) Sentence phrasing is awkward and in need of a re-write.
Response: It was reformulated.
[35] (lines 147-152) Did the 500 m above sea level air parcel trajectory ever go under the ground?
Response: It was a mistake in the text: three altitudes above ground level were applied to the HYSPLIT model (proper version in Fig. 7).
[36] (lines 154-310) Take special care to examine verb tenses in the “Results” section as they occasionally flip between present and past tense.
Response: We tried to introduce some order; some universal information are given in present tense and particular facts from past in the past tense.
[37] (line 155) should read “The mean annual precipitation total…”
[38] (line 162) Terminology “XI” and “IV” is unfamiliar. Do these refer to the months November and April? If so, please spell out the months rather than using roman numerals to represent the month of the year.
Response [37-38]: Appropriate changes were made.
[39] (line 174) The result “…and exceeding 100% in the summer months (Fig. 3)” is confusing as the median only exceeds 100% in July in Figure 3. Please clarify what is meant in this statement.
Response: The paragraph was reformulated and Figure 3 (Fig. 2 in the revised version) was described in details.
[40] (line 178) should read “Based on the methodology described in the previous section,…”
Response [40]: Corrected.
[41] (line 185) Please clarify the meaning of “…the rate of increment of the flood wave…”.
Response: The sentence in question was reformulated.
[42] (line 195) should read “Regions of heavy precipitation include the area of…”
Response [42]: Corrected.
[43] (lines 202-203) “…PW over the studied area decreases from the south to the north…” is counterintuitive and is not true over the entire region displayed in Figure 5. It is also contradicted in the statement made on lines 343-344.
Response: The wrong statement was deleted and the entire paragraph was reformulated.
[44] (line 281 and elsewhere in the manuscript) should read “…northeast…” Be careful to use ‘northeast’, ‘southeast’, ‘northwest’, and ‘northeast’ without the dash (‘-’) between the words.
Response: Corrected everywhere.
[45] (lines 221-222) the statement “…advection of humid air masses from the southeast…” contradicts the expected flow (northwest) with the cyclone center located due east of the study domain. It also contradicts most of the HYSPLIT trajectories showing the 500 m path directed from the north, northwest, or northeast displayed in Figure 7. It is also contradicted in the statement made on lines 343-344.
Response: The questioned statement seems to be wrong; therefore the doubtful sentence was deleted from results.
[46] (lines 227-229) Unusually brief (single-sentence) paragraphs should be avoided.
Response: Corrected.
[47] (lines 230-296) Please find my major concern items (A) and (B) described above.
Response: Considering (A), the number of clusters were added to the description of each type and composition of each group (type) is given in Figure A1.
Considering (B), in order to supplement our analysis with the upper-level weather patterns we added maps of anomalies of 500 hPa geopotential level (Fig. 6a,b). Please, see our detail explanation above.
[48] (line 304) Why was a 72-h backward trajectory examined (rather than a 96-h or 48-h trajectory)?
Response: We tried different periods and 3-days trajectories seemed to be the best for showing circulation in the low pressure system around the cyclonal centre.
[49] (line 316) “However, it was proved in this study…” This is an overly absolute statement given the modest analysis of the manuscript.
Response: The sentence was reformulated.
[50] (line 349) Was an examination undertaken to determine if any of the 17 case studies was influenced by an atmospheric river? If not, why not?
Response: According to Lavers and Villarini (2014, 2015) atmospheric rivers do not affect precipitation in southern Poland in summer. The landfalling atmospheric rivers have the most impact on western Europe. However, they reach as far east as to central Europe (Germany, Poland – mainly northern part) in late spring, winter and autumn. We supplemented some additional comments on this issue in the discussion. We think that influence of atmospheric rivers on precipitation and floods in Poland during entire hydrological year is an issue for further studies.
References:
Lavers, D.A.; Villarini G. The nexus between atmospheric rivers and extreme precipitation across Europe. Geophys. Res. Lett. 2014, 40, 12.
Lavers A.D.; Villarini G. The contribution of atmospheric rivers to precipitation in Europe and the United States. J. Hydrol. 2015, 522, 382–390.
also
Champion, A.J.; Allan, R.P.; Lavers, D.A. Atmospheric rivers do not explain UK summer extreme rainfall. J. Geophys. Res. Atmos. 2015, 120, 6731–6741.
[51] (lines 371-387) The ‘Conclusions’ section will need to be re-written in light of the new results created in response to my major concern item (B) described above.
Response: Conclusions were modified and some concepts for future research motivated by this study were specified.
Figures
[52] More details need to be added to the captions of all figures that specify the units of fields displayed.
Response: All captions were extended and complemented with units.
[53] Figure 1 caption must specify the number of each type of observing station used in the study.
Response: Number of each type of observing station used in the study was specified in Figure 1 caption.
[54] Figure 2 caption must specify the months of the study period (rather than using roman numerals).
Response: Roman numerals were substituted with months names in Figure 2 (Figure A1 in the revised version of the manuscript).
[55] Figure 3 must utilize the names of the months along the abscissa (x-axis).
Response: Roman numerals were substituted with months names in Figure 3 (Figure 2 in the new version).
[56] Figure 4 must be split into two parts as it is nearly impossible to read the text in the figure.
Response: Figure 4 was changed according to also other reviewers suggestions, namely number of demonstrated cased was reduced to four. Besides, fonts were enlarged and the figure quality was improved (it is Fig. 3 in the revised version of the manuscript).
[57] Figure 5 caption must be corrected to reflect that SLP is plotted in the left-hand column of the figure, while PW is plotted in the right-hand column of the figure. The caption of the figure should list the number of events contained in each of the composites. The caption of the figure needs to define the region highlighted by the red rectangle. Also, the color scale range of panel C is too large since values on the figure don’t come close to reaching -10 or +12 mm.
Response: Figure caption was corrected and the number of cases (all 17 in each composite) was specified. Region highlighted by the red rectangle was defined and the range of the color scale was diminished.
[58] Figure 6 must be split into two parts as it is nearly impossible to see the SLP anomaly contours. The caption of the figure should list the number of events contained in each of the five types.
Response: Figure 6 was split into two parts, with all maps enlarged (Figure 5a,b in the revised version).
[59] Figure 7 must define the Type category of each case. Also, the authors need to make clear why these particular nine cases were shown in the figure, when eight other cases were not plotted. The caption should also read ‘air parcels’ instead of ‘air particles’.
Response: Type category of each case was defined and comments concerning this matter were supplemented in the first paragraph of section 4.4. We thought, that showing all cases would be too much (too small maps in one figure), so 9 cases (3 by 3 in the figure) were chosen and it was rather a random choice.

Reviewer 2 Report
The paper is clear, well-structured and the subject is within the scope of the Journal.
The manuscript deals with a description of hydro meteorological condition during floods event in the area of interest analyzing 17 cases occurred between 1971-2015 subdividing it in 5 groups.
In my opinion, the drawback of the paper is the lack of novelty, the Authors in the Discussion confirm the role of orography (lines 320-322) and that low pressure systems are responsible for heavy precipitation although the different origin can play a role (lines 325-326). These are well known causes of floods not only in Polish Sudeten Mountains, as well as well-known and analyzed is the influence of the precipitation associated with Mediterranean cyclones and that Mediterranean is a source of humidity transportation.
Nonetheless, the study is worthy of being published, since the carried out analysis has a climatologic value for the area.
In the following some minor remarks:
Figure 2: In my opinion, Figure 2 is not necessary, its content is explained in the text, I suggest to delete it and, instead, to insert figure A1 in the main text, in order to maintain the same number of figures.
Figure 5 and 6: It seems to me, that the captions are missing of the reference to the red rectangle. Figure 6 is really difficult to read; I suggest to delete some plots.
Author Response
Reviewer 2
We thank very much the Reviewer for taking time and effort in reading and analyzing our manuscript entitled “Classification of synoptic conditions of Atmospheric forcing on summer floods in Polish Sudeten Mountains” (previous title “Atmospheric forcing on summer floods in Polish Sudeten Mountains”). We are very grateful for all comments, which helped to improve the manuscript. We provided a record of all changes introduced to the manuscript, which were suggested by all Reviewers. Changes are marked in the annotated version of the manuscript enclosed as a supplementary material (.pdf).
Attached below, please find our point-by-point responses to all of your comments.
Sincerely,
Ewa Bednorz and Co-Authors
Comments and Suggestions for Authors
In my opinion, the drawback of the paper is the lack of novelty, the Authors in the Discussion confirm the role of orography (lines 320-322) and that low pressure systems are responsible for heavy precipitation although the different origin can play a role (lines 325-326). These are well known causes of floods not only in Polish Sudeten Mountains, as well as well-known and analyzed is the influence of the precipitation associated with Mediterranean cyclones and that Mediterranean is a source of humidity transportation.
Nonetheless, the study is worthy of being published, since the carried out analysis has a climatologic value for the area.
Response: Although, the causes of floods and extreme precipitation in Polish mountains, and the role of Mediterranean cyclones was described in many works, we tried to complement this knowledge with the description of tracks of cyclones causing floods in the Sudeten. This was done by the analysis of synoptic conditions preceding each flood case. We perceive some novelty in such approach. Besides, the analysis of precipitable water content during floods and before them, gives some new information on the moisture transport and on causes of floods in the Sudeten.
Minor remarks:
Figure 2: In my opinion, Figure 2 is not necessary, its content is explained in the text, I suggest to delete it and, instead, to insert figure A1 in the main text, in order to maintain the same number of figures.
Response: Figure 2 was removed from the main text to the appendix (Fig. A2 in the revised version).
Figure 5 and 6: It seems to me, that the captions are missing of the reference to the red rectangle. Figure 6 is really difficult to read; I suggest to delete some plots.
Response: Captions of Figure 5 (Fig. 4 in the revised version) and Figure 6 (Fig. 5a,b in the revised version) were supplemented and red rectangles explained. Figure 6, which was difficult to read indeed, was divided into two parts with all maps enlarged (Fig. 5a,b).

Reviewer 3 Report
Please see attachment.

Author Response
Reviewer 3
We thank very much the Reviewer for taking time and effort in reading and analyzing our manuscript entitled “Classification of synoptic conditions of Atmospheric forcing on summer floods in Polish Sudeten Mountains” (previous title “Atmospheric forcing on summer floods in Polish Sudeten Mountains”). We are very grateful for all of the valuable comments, both general and detailed ones, which allowed to improve the manuscript. All of comments and remarks were considered carefully; the general comments allowed to revise our ideas concerning the studied issue and they are essential for our future studies.
We provided a record of changes introduced to the manuscript, according to all Reviewers’ suggestions. Changes are marked in the annotated version of the manuscript enclosed as a supplementary material (.pdf).
Attached below, please find our point-by-point responses to all of your comments.
Sincerely,
Ewa Bednorz and Co-Authors
Major Comments:
1. The authors tried to link the sea level pressure to the flooding. While I agree that high precipitation tends to be associated with cyclones so that the sea level pressure, the direct connection to the flooding may require other conditions.
Response: We supplemented the study with the analysis of pressure conditions in the middle troposphere, i.e., anomalies of the geopotential level of 500 hPa.
2. Why do you use PW, not IVT (integrated vapor transport)? The authors use precipitation water as the precursor of the heavy precipitation and flooding associated with cyclones and accompanying front. The moisture transport by the wind can be a more critical factor than the moisture itself to cause heavy precipitation. This paper showed PW and described it as “moisture transport” but this reviewer disagrees with that. Instead of PW, water vapor fluxes should be taken into account. These moisture transport by the wind can be simply calculated by the multiplication of the specific humidity and the wind vector and integrate them over the entire atmosphere (at least 10 km, top of the troposphere, e.g. Neiman et al. 2008). The more than 70% of the water vapor is known to reside within 3 km about the surface, so IVT should be at least integrated from the earth surface to 3km and preferably higher.
Response: Precipitable water is a commonly used variable and it gives a total content of water in the troposphere. We understand that IVT give also an information on moisture transport, however having a sequence of PW maps (6, 4, 2 days before floods and the “0” day) we can see the changes and thereby general direction of moisture transport over Europe.
3. While Figure 6 could be the compelling figures and seems to be one of the most important figures, but this is actually not very visible.
Response: Figure 6 (Figure 5a,b in the revised version) was divided into two parts, with all maps enlarged.
4. The authors classified the synoptic circulation patterns to cause heavy precipitation and flooding into five types. Why do you group the circulations into five types? Please explain further why you categorized them into 5 types.
Response: Figure A1, showing results of hierarchical clustering by the Ward’s method, with the composition of each group, was added in the Appendix (Fig. A1) and commented in the text. The Ward method was described in details in section 3. Considering the number of clusters, the principal goal is to find that level of clustering that maximizes similarity within clusters and minimizes similarity between clusters, however in practice the best number of clusters is usually not obvious. Wilks (2011) postulates that determining the number of groups requires a subjective choice that depends on the goals of the analysis. We intended to obtain not too big and possibly equally numbered groups and not too many of them, considering that we have 17 object to classify. As can be seen in the Figure A1 a larger number of groups was rather not possible. Establishing 4 groups would impel to join type 4 and 5 which differ significantly, the same problem with dividing into 3 groups.
Reference: Wilks, D. Statistical methods in the atmospheric sciences. Elsevier Academic Press, 2011, Amsterdam
5. As a classification method, the author chose Ward’s method. While it was mentioned by the authors that this is a well known method in the climate study, this reviewer was not very familiar with this method. Can this method be explained in more detailed manner and demonstrated it compared to Self-Organizing map (SOM) or K-mean clustering method?
Response: A broader description Ward method was introduced in the manuscript. The dendrogram showing results of hierarchical clustering by the Ward’s method, with indicated composition of each group was added in the Appendix (Fig. A1) and commented in the text. The Ward method was described in details in section 3.
Additionally, we compared the results of clustering by Ward’s method with K-means:
Ward K-means
1. 1997(2), 2009, 2010(2) | 1997(2), 2009, 2010(2)
2. 1977(2), 1980, 2001 | 1977(2), 1980, 2001
3. 1975, 1977, 1997, 2006 | 1975, 1977, 1997, 2006
4. 1985, 2002, 2010, 2011 | 1985, 2002, 2010
5. 1971, 1981, 2013 | 1971, 1981, 2011, 2013
Only flood from 2011 changed its position. We checked the archive reanalysis (available at https://www.wetterzentrale.de/reanalysis) and found rather western cyclones before 22 Jul 2011, so it seems flood of 2011 should rather belong to type 4 (as it is in the manuscript).
6. The factors to control the synoptic condition and atmospheric circulations can be more than two, just SLP and PW (this can be related to temperature too). Have you considered looking at another factor to control heavy precipitation, such as wind?
Response: We realize that there are much more factors controlling synoptic conditions of occurrence of extreme precipitation, e.g., air temperature in different tropospheric levels, relative humidity, potential vorticity and wind is also one of them. However, we tried to concentrate on pressure conditions, which directly induce particular air masses (and moisture) transport. We mentioned new concepts for future research in the last paragraph of conclusions.
7. The contribution of each type to the flood culmination was not clearly stated. How much does each type contribute to the actual flood (e.g., type 1 contribute 15% to flooding, and type 2 contribute 25% etc)? Are they the same contribution before and on that day? Providing the relative contribution of SLP and PW to the actual flood will strengthen the paper. Maybe you can try to make it as a table.
Response: Unfortunately, we cannot quite understand the idea of ‘relative contribution’ and how to compute it.
8. This reviewer had a hard time to understand and interprete Fig. 7. What are the main points of Fig 7? The purpose of this may be to show the different source and origin of air mass affecting flooding in Polish mountine regions, but how are they related to the specific types shown in Fig. 6? Or, are they to show the different trajectory patterns and to explain that they are associated with a different location, intensity, and pattern of the existing SLP and PW?
Response: Specification of the type represented by each map of trajectories was indeed necessary, and it was supplemented to each map in Figure 7. We also extended the description of backward trajectories in section 4.2, mainly in context of represented type.
Minor Comments.
1. I recommend changing the title. The authors just focus on the sea level pressure and the precipitable water, but they regard it as the “atmospheric forcing”, I think the more appropriate title can be “ Classification of Synoptic conditions on summer floods on summer floods in Polish Sudeten Mountains”
Response: The title was changed according to suggestion – we also think it is more appropriate.
2. Line 375-376: This is related to one of major comments mentioned above. “The evolution of the pressure field and also the moisture transport was investigated, using the anomaly-based method”. It is mentioned. However, the authors only examined the precipitable water, but the actual moisture transport. In order to take into account the moisture transport, not only moisture but also the “wind’ carrying the moisture should be considered.
Response: The sentence in question was reformulated – we meant the moisture content.
3. The authors should have looked at the other meteorological condition, such as wind, lower-level, and even upper-level jet and its relation to the formation and development of the sea level pressure if the purpose of this paper is to describe atmospheric forcings on the flooding events . The authors may consider changing the title of the manuscript.
Response: : In order to supplement our analysis with the upper-level weather patterns we added maps of anomalies of 500 hPa geopotential level (Fig. 6a,b). Composite anomaly maps constructed for each type picture meridional flow configuration in the middle troposphere during extreme precipitation events. The title was changed according to above suggestion.
4. Some figures are not easily readable so hard to interpret the figures (e.g., Fig. 4, Fig.6).
Response: The quality of most figures was improved (including Fig. 4 and 6, which are numbered 3 and 5a,b in the revised version of manuscript).
5. The figure captions need to be revised and more specific (e.g., Figs. 3-4)
Response: All figure caption were complemented and made more specific.

Reviewer 4 Report
The paper is devoted to the analysis of typical synoptic situations preceding selected flood cases in the Sudeten mountains. The paper has convincible regional importance. Its results are based on sufficient data supply. The disadvantage is that illustration and description of some results do not allow to fully assess the quality of the paper. Generally more attention to classification on types should be payed, the representation and analysis of the Figures must be improved.
Specific comments:
How many and which flood cases are there in the groups? It is important to provide selected years of flood cases and their division on groups (line 178). Procedure and results of classification on flood types must be better described in Data and methods and further in the text (line 234). There is an impression from the Fig. 6 that the main difference of types is only in cyclone tracks. How the types identified correspond to the earlier papers, e.g. by Wypych et al., 2018 [20] and others. Can these types be generalized?
I recommend to choose keywords different from the title and abstract for higher visibility of a paper.
Specify the months of the summer half-year in the text (line 156).
Representation (on the Figure 1) of the geographical objects mentioned in the text is needed (mountains and chosen rivers). Otherwise generalize geographical (spatial) description of objects in the text (e.g. section 4.1).
I suppose it is better to highlight isolines on the Fig. 2 and Fig. 5.
Describe shortly Parde’s coefficient meaning and /or formula (line 173).
The reason of selection of floods in Fig. 4 is not mentioned. There are similar graphs there so as on Fig. 7 which are scarcely described in the text. I suggest to illustrate only typical examples and to provide mentioned on Fig. 4 geographical locations on Fig. 1 (lines 189-196).
Section 4.2 is not only about SLP but also PW (line 197).
New paragraph is better to begin on line 202 and to remind which studied area is implied.
There is inappropriate context for reference on Fig. 5 on line 202.
Logical construction of paragraph (sequence of sentences) on lines 208-211 regarding “modifications” is to be provided.
Reference on the Fig. 5 is required on line 215.
Specify why you make such a conclusion and how the direction of inflow if humid air masses can be seen, especially from the south-east (lines 221-223).
“Left” and “right” are mixed on line 225.
Check grammar on line 232.
Fig. 6 must be improved in terms of visualization of results. SLP positive and negative isolines cannot be seen properly. Column A (6 days before) seems an extra one.
There are missing “SLP anomalies” on line 251 and “high SLP center”on line 262.
Check description of movement of humid air masses from the southern sector for every type (and specify of which region), especially in type 3 for which southern inflow is stated in a row with western on lines 272-273. Also check where you locate blockage anticyclone which should last for several days. According to the Fig. 6 (D, type 5) the highest PW anomalies are on the culmination day (line 296).
Reference on the Fig. 7 is needed in the text on lines 305-307.
Better description is needed in the text how it is proved that meteorological conditions are the most important factors… (line 316-317).
Try to avoid identical sentences in the Abstract, Introduction and Conclusions sections and pay enough attention to each conclusion you make in the description of results.
Author Response
Reviewer 4
We thank very much the Reviewer for taking time and effort in reading and analyzing our manuscript entitled “Classification of synoptic conditions of Atmospheric forcing on summer floods in Polish Sudeten Mountains” (previous title “Atmospheric forcing on summer floods in Polish Sudeten Mountains”). We are very grateful for all valuable comments, which helped to improve the manuscript.
We provided a record of all changes introduced to the manuscript. Changes are marked in the annotated version of the manuscript enclosed as a supplementary material (.pdf).
Attached below, please find our point-by-point responses to all of your comments.
Sincerely,
Ewa Bednorz and Co-Authors
Comments and Suggestions for Authors
· The disadvantage is that illustration and description of some results do not allow to fully assess the quality of the paper.
Response: We improved the quality of all figures and we tried to complement the descriptions.
· Generally more attention to classification on types should be payed, the representation and analysis of the Figures must be improved.
Response: The Ward method was described in details in section 3 and the extended comments on results of classification were added in section 4.3.
Specific comments:
· How many and which flood cases are there in the groups? It is important to provide selected years of flood cases and their division on groups (line 178). Procedure and results of classification on flood types must be better described in Data and methods and further in the text (line 234). There is an impression from the Fig. 6 that the main difference of types is only in cyclone tracks. How the types identified correspond to the earlier papers, e.g. by Wypych et al., 2018 [20] and others. Can these types be generalized?
Response: Figure A1, showing results of hierarchical clustering by the Ward’s method, with the composition of each group was added in the Appendix and commented in the text. The Ward method was described in details in section 3. Also Table A1 with characteristic features of each flood (and belonging to a particular type) was introduced. The impression that the main difference of types is in cyclone tracks is reasonable as it was the main aim of the division – storm tracks determination (emphasized now in last paragraph of the first section and explained in the methods).
Wypych et al. (2018) and Niedźwiedź et al. (2015) found cyclonic circulation types with inflow from the northern sector in low troposphere, responsible for heavy precipitation in mountainous southern Poland. This corresponds well with our findings. Niedźwiedź et al. postulated that these cyclonic situations are associated with cyclones following track Vb after van Bebber. We found also western cyclone tracks being responsible for flood-bringing precipitation in the Polish Sudeten.
Generalization of the division into 5 groups would lead to loosing information. For example, establishing 4 groups would impel to join type 4 and 5 which differ significantly, the same problem with dividing into 3 groups (please see Fig. A1 and Fig 5a,b).
The main difference between type are cyclone tracks, indeed – this was the main purpose of the division and it comes from the data used for the clustering, i.e., the standardized daily SLP anomalies in the days preceding floods.
· I recommend to choose keywords different from the title and abstract for higher visibility of a paper.
Response: Keywords were changed.
· Specify the months of the summer half-year in the text (line 156).
Response: Complemented.
· Representation (on the Figure 1) of the geographical objects mentioned in the text is needed (mountains and chosen rivers). Otherwise generalize geographical (spatial) description of objects in the text (e.g. section 4.1).
Response: Geographical names mentioned in the text were introduced to Figure 1.
· I suppose it is better to highlight isolines on the Fig. 2 and Fig. 5.
Response: Both figures were improved.
· Describe shortly Parde’s coefficient meaning and /or formula (line 173).
Response: The formula of Parde’s coefficient was added in the text, together with appropriate explanations.
· The reason of selection of floods in Fig. 4 is not mentioned. There are similar graphs there so as on Fig. 7 which are scarcely described in the text. I suggest to illustrate only typical examples and to provide mentioned on Fig. 4 geographical locations on Fig. 1 (lines 189-196).
Response: Figure 4 (Fig. 3 in the revised version of the manuscript) was changed and only four examples from the highest flood in July 1997 were demonstrated.
· Section 4.2 is not only about SLP but also PW (line 197).
Response: Title of section 4.2 was reformulated.
· New paragraph is better to begin on line 202 and to remind which studied area is implied.
Response: Suggested changes were made.
· There is inappropriate context for reference on Fig. 5 on line 202.
Response: Reference on Fig. 5 (Fig. 4 in the revised version) was moved farther.
· Logical construction of paragraph (sequence of sentences) on lines 208-211 regarding “modifications” is to be provided.
Response: The first sentence in paragraph was reformulated.
· Reference on the Fig. 5 is required on line 215.
Response: Reference was introduced (on Fig. 4 in a revised version).
· Specify why you make such a conclusion and how the direction of inflow if humid air masses can be seen, especially from the south-east (lines 221-223).
Response: The conclusion was wrong, so the sentence in question was deleted.
· “Left” and “right” are mixed on line 225.
Response: Corrected.
· Check grammar on line 232.
Response: Corrected.
· Fig. 6 must be improved in terms of visualization of results. SLP positive and negative isolines cannot be seen properly. Column A (6 days before) seems an extra one.
Response: Figure 6, which was difficult to read indeed was divided into two parts with all maps enlarged (Fig. 5a and 5b in a revised version of the manuscript). Column A is spare for types 1, 3 and 5, in types 2 and 4 a cyclonal center starts to form 6 days before flood culmination.
· There are missing “SLP anomalies” on line 251 and “high SLP center”on line 262.
Response: Corrected.
· Check description of movement of humid air masses from the southern sector for every type (and specify of which region), especially in type 3 for which southern inflow is stated in a row with western on lines 272-273. Also check where you locate blockage anticyclone which should last for several days. According to the Fig. 6 (D, type 5) the highest PW anomalies are on the culmination day (line 296).
Response: On the description of type 3 (and others) we first describe the movement of the SLP negative anomaly center (which mean in fact cyclonic center) from the west in type 3, and at the same time we mention the source region of the air masses coming from the south – according to the counter clockwise circulation around the cyclonic centre. We rather do not see a contradiction in it. We checked location of the positive SLP anomalies in type 3, which make blockage and they spread as a ridge from the Athletic (west) to the Scandinavia (north).
We meant the highest PW anomalies over the studied area – Polish Sudeten. It has been precised in the text.
· Reference on the Fig. 7 is needed in the text on lines 305-307.
Response: Reference was added.
· Better description is needed in the text how it is proved that meteorological conditions are the most important factors… (line 316-317).
Response: It was too absolute statement, indeed. We reformulated the sentence.
· Try to avoid identical sentences in the Abstract, Introduction and Conclusions sections and pay enough attention to each conclusion you make in the description of results.
Response: We tried to reformulate all mentioned section, paying more attention to conclusions.

Reviewer 5 Report
General Comments
This manuscript describes a study examining synoptic conditions related to summer floods in the Polish part of the Sudeten mountains. This is a relevant topic and the methodology used by the authors is adequate to the problem (anomalies analysis of sea level pressure and precipitable water, and Hysplit-derived retrotrajectories assessment). However, some parts of the manuscript must be corrected (for example precipitable water should not be introduced as an exotic or 'very complex index') or simply better described (i.e., how do authors decide which are the 5 synoptic groups patterns). Any way I found the results worth publishing: for example they determine that approximately northern (N or NE) flow impinging over the area of study dominate during the flooding period (strong and moist flow perpendicular to an approximately linear mountain massif). This is consistent with previous studies performed elsewhere with observational field campaigns or numerical higher resolution analysis such as Medina et al. (2005) or Trapero et al (2013) which I think should be commented to put the results in a wider context. The manuscript also contains a number of formal issues that should be solved. In summary, for all the above I encourage authors to consider these items and specific comments below to improve their manuscript for a future submission to improve their manuscript in a corrected version.
Specific comments
Page 1, line 12, please check English: Polish the Sudeten -> the Polish Sudeten
Page 1-2, line 44-45, please check English: from an extremely high rainfall rates -> from extremely high rainfall rates ? [check original meaning]
Line 51, comments about ideal conditions for orographic precipitation enhancement could be included here - (Medina et al. (2005) or Trapero et al (2013), see references below - and commented in the discussions that results obtained by the authors are consistent with them.
Line 63, please indicate here the exact number of flood case studies examined.
Section 2. You mention here (and elsewhere in the manuscript) several locations (rivers, mountains, etc.) which I could not locate in the map so the text cannot be followed unless the reader is familiar with the region of study. As this is an international journal intended for a global audience you should add a label of each local name cited to Figure 1.
Page 3, figure 1. A part from the previous comment I have two additional suggestions for this figure. Firstly, please consider adding another (perhaps smaller) map showing the region of study in a broader context (such as Central Europe). Secondly, the labels of the locations are too small to be read, please enlarge them.
Line 105, typo (I presume): weakly -> weekly
Line 107, please check English: which domain -> whose domain
Lines 120-121. Content problem. Precipitable water is a fundamental and basic concept which I do not think should be introduced as "a kind of complex index" in an international journal as Water. I think it is ok if you want to give the definition, you can simply write: Precipitable water, which can be defined as 'the total...'.
Page 4, line 125, similarly as the previous comment: PW is a complex index that expresses -> PW is a variable that expresses
Line 129. Content problem. The spatial resolution you use is not adequate to identify most mesoscale patterns (for example you are going to miss a lot of secondary lows) so you should change here: mesoscale -> synoptic
Page 4, line 162. Use of Roman numbers to indicate months should be explained to avoid confusions.
Page 5, either in section 4.2 or perhaps in the appendix, you should add a table listing details of each of the 17 flood cases, including at least: initial and end date, maximum precipitation during the period (and station), maximum 24h precipitation (and station). I would also indicate the type of SLP pattern (1 to 5) in which you classify the events.
Page 6, please enlarge fonts: they cannot be read now (perhaps simply make the figure larger, a 1-page figure).
Line 195, English: ammount -> amount
Sub-section 4.2 should be 4.3 (there is already a sub-section 4.2 in the previous page).
Line 197, suggest: Field of the sea level -> Sea level
Line 201, English: occurres -> occurs
Lines 204. Content problem: note the current sentence is not correct; because amount of PW -> because the maximum amount of PW
Lines 204. Content problem - similarly as before. more water -> more water vapour
Line 208 English: are -> is
Page 7, . figure 5 caption. The right and left columns content description are swapped (i.e. SLP left) - please correct.
Line 229, English, suggest: system -> systems [please check meaning]
Page 7, last paragraph. Authors do not explain here how they decide that there are 5 different SLP patterns. Is it a manual, subjective decision? Please clarify. You should also indicate how many flood events are used for each type to have an idea of how frequent or representative each type is - note that if you included this information in Table 1 requested above you can simply reference it here.
Line 232, English, had were -> had
Line 233, suggest: different location and extent, various intensity -> different location, extent, and intensity
Page 8, line 279, English: comparison -> compared
Page 9, Figure 6. Please enlarge, as current panels cannot be seen correctly (perhaps a one page figure would solve this). Maybe also considering vertically aligned labels of "Type 1", "Type 2", etc. will help to make a more clear plot (wider space for the panels).
Line 301, again numbering problem. section 4.2: now should be 4.4
Page 9, last paragraph: why are only 9 cases selected from the 17 considered? Are they representative for the different Types mentioned earlier? Please comment briefly in the text.
Page 10. Numbering problem. Section 4 should be Section 5. Please check and correct.
Page 11. Numbering problem. Section 5 should be Section 6. Please check and correct.
References section. You cite a number of local publications in Polish which might be potentially interesting for readers who do not speak this language. I suggest you translate the original title of each study to English and the indicate "(In Polish)" so interested readers at least know their existence.
REFERENCES
Medina S, Smull BF, Houze Jr RA, Steiner M (2005): Cross-barrier flow during orographic precipitation events: Results from MAP and IMPROVE. Journal of the Atmospheric Sciences, 62(10), 3580-3598.
Trapero L, Bech J, Duffourg F, Esteban P, Lorente J (2013): Mesoscale numerical analysis of the historical November 1982 heavy precipitation event over Andorra (Eastern Pyrenees). Natural Hazards and Earth System Sciences, 13, 2969-2990
Author Response
Reviewer 5
We thank very much the Reviewer for taking time and effort in reading and analyzing our manuscript entitled “Classification of synoptic conditions of Atmospheric forcing on summer floods in Polish Sudeten Mountains” (previous title “Atmospheric forcing on summer floods in Polish Sudeten Mountains”). We are very grateful for all valuable comments, which helped to improve the manuscript. We provided a record of all changes introduced to the manuscript (suggested by all Reviewers). Changes are marked in the annotated version of the manuscript enclosed as a supplementary material (.pdf).
Attached below, please find our point-by-point responses to all of your comments.
Sincerely,
Ewa Bednorz and Co-Authors
General Comments
· However, some parts of the manuscript must be corrected (for example precipitable water should not be introduced as an exotic or 'very complex index') or simply better described (i.e., how do authors decide which are the 5 synoptic groups patterns).
Response: Description of precipitable water was reformulated (definition from Glossary of American Meteorological Society) and equation added, according to suggestions of two reviewers.
A broader description of Ward method was introduced in the manuscript. Figure A1, showing results of hierarchical clustering by the Ward’s method, with the composition of each group was added in the Appendix and commented in the text. The Ward method was described in details in section 3. Also Table A1 with characteristic features of each flood (and belonging to a particular type) was introduced.
· This is consistent with previous studies performed elsewhere with observational field campaigns or numerical higher resolution analysis such as Medina et al. (2005) or Trapero et al (2013) which I think should be commented to put the results in a wider context.
Response: Synoptic patterns and mesoscale mechanisms for orographic precipitation enhancement described in Medina et al. (2005) and Trapero et al (2013) were mentioned in the discussion, both papers were cited.
Specific comments
Page 1, line 12, please check English: Polish the Sudeten -> the Polish Sudeten
Response: Corrected.
Page 1-2, line 44-45, please check English: from an extremely high rainfall rates -> from extremely high rainfall rates ? [check original meaning]
Response: Changed to “from extremely high rainfall rates”.
Line 51, comments about ideal conditions for orographic precipitation enhancement could be included here - (Medina et al. (2005) or Trapero et al (2013), see references below - and commented in the discussions that results obtained by the authors are consistent with them.
Response: Mesoscale mechanisms for orographic precipitation enhancement recognized by Medina et al. (2005) and Trapero et al (2013) were mentioned in the discussion, as well as the similarity of synoptic conditions of extreme precipitation recognized in our manuscript and by Medina et al. (2005) and Trapero et al (2013), i.e., the strong cyclonic flow induced by the deep lows.
Line 63, please indicate here the exact number of flood case studies examined.
Response: Number of floods (17) was introduced.
Section 2. You mention here (and elsewhere in the manuscript) several locations (rivers, mountains, etc.) which I could not locate in the map so the text cannot be followed unless the reader is familiar with the region of study. As this is an international journal intended for a global audience you should add a label of each local name cited to Figure 1.
Response: Geographical names mentioned in the text were introduced in Figure 1.
Page 3, figure 1. A part from the previous comment I have two additional suggestions for this figure. Firstly, please consider adding another (perhaps smaller) map showing the region of study in a broader context (such as Central Europe). Secondly, the labels of the locations are too small to be read, please enlarge them.
Response: The location of Polish Sudeten in a broader context is given as red rectangles in Figures 4, 5 and 6 (according to new numbering in the revised version). The meaning of the red rectangles, not explained before, is now entered in the captions of Figures 5 and 6 (now Figures 4 and 5a,b in the revised version of the manuscript).
Line 105, typo (I presume): weakly -> weekly
Response: Corrected.
Line 107, please check English: which domain -> whose domain
Response: Corrected.
Lines 120-121. Content problem. Precipitable water is a fundamental and basic concept which I do not think should be introduced as "a kind of complex index" in an international journal as Water. I think it is ok if you want to give the definition, you can simply write: Precipitable water, which can be defined as 'the total...'.
Response: Description of precipitable water was reformulated (using the definition from Glossary of American Meteorological Society) and equation added, according to suggestions of two reviewers.
Page 4, line 125, similarly as the previous comment: PW is a complex index that expresses -> PW is a variable that expresses
Response: Corrected.
Line 129. Content problem. The spatial resolution you use is not adequate to identify most mesoscale patterns (for example you are going to miss a lot of secondary lows) so you should change here: mesoscale -> synoptic
Response: The sentence was reformulated according to the Reviewer’s comment.
Page 4, line 162. Use of Roman numbers to indicate months should be explained to avoid confusions.
Response: Roman numbers were changed to months’ names everywhere in the manuscript.
Page 5, either in section 4.2 or perhaps in the appendix, you should add a table listing details of each of the 17 flood cases, including at least: initial and end date, maximum precipitation during the period (and station), maximum 24h precipitation (and station). I would also indicate the type of SLP pattern (1 to 5) in which you classify the events.
Response: A suggested table with some basic characteristics of floods and their belonging to the distinguished type was added to the appendix (Table A1).
Page 6, please enlarge fonts: they cannot be read now (perhaps simply make the figure larger, a 1-page figure).
Response: Figure 4 was changed according to also other reviewers suggestions, namely number of demonstrated cased was reduced to four. Besides, fonts were enlarged and the figure quality was improved (it is Fig. 3 now in the revised version of the manuscript).
Line 195, English: ammount -> amount
Response: Corrected.
Sub-section 4.2 should be 4.3 (there is already a sub-section 4.2 in the previous page).
Response: All numbering of sections and subsections were corrected.
Line 197, suggest: Field of the sea level -> Sea level
Response: The title of subchapter was reformulated.
Line 201, English: occurres -> occurs
Response: Corrected.
Lines 204. Content problem: note the current sentence is not correct; because amount of PW -> because the maximum amount of PW
Response: The entire paragraph in section 4.3. concerning amount of PW was reformulated.
Lines 204. Content problem - similarly as before. more water -> more water vapour
Response: Corrected.
Line 208 English: are -> is
Response: Corrected.
Page 7, . figure 5 caption. The right and left columns content description are swapped (i.e. SLP left) - please correct.
Response: Corrected.
Line 229, English, suggest: system -> systems [please check meaning]
Response: Corrected.
Page 7, last paragraph. Authors do not explain here how they decide that there are 5 different SLP patterns. Is it a manual, subjective decision? Please clarify. You should also indicate how many flood events are used for each type to have an idea of how frequent or representative each type is - note that if you included this information in Table 1 requested above you can simply reference it here.
Response: The tree diagram, which is a graphical picture of Ward method was added to the appendix (Figure A1) with explanation in the text, also in Table A1 the number of type was assigned to each flood (reference to table added in the text).
Line 232, English, had were -> had
Response: Corrected.
Line 233, suggest: different location and extent, various intensity -> different location, extent, and intensity
Response: Corrected.
Page 8, line 279, English: comparison -> compared
Response: Corrected.
Page 9, Figure 6. Please enlarge, as current panels cannot be seen correctly (perhaps a one page figure would solve this). Maybe also considering vertically aligned labels of "Type 1", "Type 2", etc. will help to make a more clear plot (wider space for the panels).
Response: Figure 6 was divided into two parts with all maps enlarged and types marked (Fig. 5a and 5b in the revised version of manuscript).
Line 301, again numbering problem. section 4.2: now should be 4.4
Response: Corrected.
Page 9, last paragraph: why are only 9 cases selected from the 17 considered? Are they representative for the different Types mentioned earlier? Please comment briefly in the text.
Response: We thought, that all cases would be too much (too small maps in one figure), so 9 cases (3 by 3 in the figure) were chosen and it was rather a random choice.
A type affiliation was added to each case in Figure 7. Besides, comments concerning this matter were supplemented in section 4.4.
Page 10. Numbering problem. Section 4 should be Section 5. Please check and correct.
Page 11. Numbering problem. Section 5 should be Section 6. Please check and correct.
Response: Corrected.
References section. You cite a number of local publications in Polish which might be potentially interesting for readers who do not speak this language. I suggest you translate the original title of each study to English and the indicate "(In Polish)" so interested readers at least know their existence.
Response: Polish titles were translated to English and introduced to citations with an annotation ‘(in Polish)’.

Round 2
Reviewer 1 Report
Minor revision suggestions...
[1] extraneous character next to "5%" (line 78) must be removed,
[2] density is a function of pressure and should be placed inside the integral in the precipitable water vapor equation (lines 129-131)
[3] Figure 6 would be much improved if regions of negative and positive 500 hPa level geopotential height anomalies were contoured differently (e.g., dashed or solid) or shaded differently (e.g., orange or blue).
[4] "Moore" is misspelled as "More" (line 434)
Author Response
We thank very much the Reviewer for taking time and effort in reading for the second time our manuscript entitled “Classification of synoptic conditions of Atmospheric forcing on summer floods in Polish Sudeten Mountains” (previous title “Atmospheric forcing on summer floods in Polish Sudeten Mountains”). We are very grateful for the kindly approving our revised version of the manuscript and for the detailed comments, which allowed to eliminate mistakes and improve the manuscript.
Attached below, please find our point-by-point responses to your comments.
Sincerely,
Ewa Bednorz and Co-Authors
Minor revision suggestions:
[1] extraneous character next to "5%" (line 78) must be removed
Response: “5‰” is correct and it means “0.5%” (‰ - per mille)
[2] density is a function of pressure and should be placed inside the integral in the precipitable water vapor equation (lines 129-131)
Response: Please, note that equation expresses precipitable water contained in a homogenous layer between levels p1 and p2. The equation as well as the definition of precipirable water was derived from the American Meterorological Society Glossary of Meteorology. We would rather leave equation as it is.
[3] Figure 6 would be much improved if regions of negative and positive 500 hPa level geopotential height anomalies were contoured differently (e.g., dashed or solid) or shaded differently (e.g., orange or blue).
Response: Figure 6 was replaced by an improved version.
[4] "Moore" is misspelled as "More" (line 434)
Response: Corrected.
Reviewer 3 Report
I think this manuscript can be published in Water at this form.Author Response
We thank very much the Reviewer for taking time and effort in reading for the second time our manuscript entitled “Classification of synoptic conditions of Atmospheric forcing on summer floods in Polish Sudeten Mountains” (previous title “Atmospheric forcing on summer floods in Polish Sudeten Mountains”).
Reviewer 4 Report
To my mind the manuscript has higher ratings after corrections. Almost all my comments were taken into account. I can argue only about the number of types which seems to me too detailed. Taking into account Fig. A1, I would rather distinguished the main types and divided them into subtypes. But of course it is always a subjective choice. Excessive detailing applies also to the use of geographical names, for example such as Jizera Mountains and the Śnieżnik Massif, etc. which are hard to perceive without mapping. Finally I advise the authors to check spelling because I have noticed some typos (e.g. in the headings) although I didn't read very intently.
Author Response
Reviewer 4 (round 2)
We thank very much the Reviewer for taking time and effort in reading for the second time our manuscript entitled “Classification of synoptic conditions of Atmospheric forcing on summer floods in Polish Sudeten Mountains” (previous title “Atmospheric forcing on summer floods in Polish Sudeten Mountains”). We are very grateful for the kindly approving our revised version of the manuscript and for the detailed comments, which allowed to eliminate mistakes.
We provided a record of changes introduced to the manuscript, according to both Reviewers’ suggestions. Changes are marked in the annotated version of the manuscript enclosed as a supplementary material (.pdf).
Attached below, please find our point-by-point responses to your comments.
Sincerely,
Ewa Bednorz and Co-Authors
· I can argue only about the number of types which seems to me too detailed. Taking into account Fig. A1, I would rather distinguished the main types and divided them into subtypes. But of course it is always a subjective choice.
Response: Initially, we intended to divide into three groups, but after producing maps for smaller groups, which differed substantially, we decided to stay with five types. The division into main types could be added to the manuscript, but it would mean introducing two new figures, which would join different cyclone tracks (type 1+2 and 4+5). We think it would rather complicate and becloud the results.
· Excessive detailing applies also to the use of geographical names, for example such as Jizera Mountains and the Śnieżnik Massif, etc. which are hard to perceive without mapping.
Response: Polish Sudeten Mountains are small, but diversified in the terms of landforms and climate, therefore a few sentences on precipitation variability within the range seems to be needed and using detail geographical names is therefore unavoidable. We tried to put all geographical names used in the text in Figure 1.; both names of mountain ranges (Jizera Mountains and the Śnieżnik Massif) mentioned by the Reviewer were also added in Fig. 1.
· Finally I advise the authors to check spelling because I have noticed some typos (e.g. in the headings) although I didn't read very intently.
Response: We checked the text again and tried to eliminate mistakes.

Reviewer 5 Report
Manuscript Number: water-523506
Title: Atmospheric forcing on summer floods in Polish Sudeten Mountains
General Comments
Authors have improved a lot the manuscript and it can be accepted for publication. There are a few minor issues, mostly related to English, that in my opinion can be checked/corrected during the production process (see below).
Specific Comments
Line 154, suggest: the quality control -> quality control
Lines 206-207, check meaning: in the decision which -> to decide which
Line 304, typo: hours. -> hours
Figure captions 5a, 5b, 6a, 6b (and similarly 7): For distinguished types -> Circulation types [In Figure caption 7, distinguished types -> circulation types]
In Table A1, according to Table title, daily (24h) rainfall amounts are listed, including 3 events exceeding 300 mm. Are they really daily or total event rainfall amounts? Fig. A3 seems to suggest that they are total event amounts, so please check this and correct if necessary.
Author Response
Reviewer 5 (round 2)
We thank very much the Reviewer for taking time and effort in reading for the second time our manuscript entitled “Classification of synoptic conditions of Atmospheric forcing on summer floods in Polish Sudeten Mountains” (previous title “Atmospheric forcing on summer floods in Polish Sudeten Mountains”). We are very grateful for the kindly approving our revised version of the manuscript and for the detailed comments, which allowed to eliminate mistakes.
We provided a record of changes introduced to the manuscript, according to both Reviewers’ suggestions. Changes are marked in the annotated version of the manuscript enclosed as a supplementary material (.pdf).
Attached below, please find our point-by-point responses to your comments.
Sincerely,
Ewa Bednorz and Co-Authors
Specific Comments
Line 154, suggest: the quality control -> quality control
Lines 206-207, check meaning: in the decision which -> to decide which
Line 304, typo: hours. -> hours
Figure captions 5a, 5b, 6a, 6b (and similarly 7): For distinguished types -> Circulation types [In Figure caption 7, distinguished types -> circulation types]
Response: All corrections were made.
In Table A1, according to Table title, daily (24h) rainfall amounts are listed, including 3 events exceeding 300 mm. Are they really daily or total event rainfall amounts? Fig. A3 seems to suggest that they are total event amounts, so please check this and correct if necessary.
Response: These are weekly precipitation totals in Table A1; the mistake was corrected.
